



# Ground-based Tropospheric Ozone Measurements: Regional tropospheric ozone column trends from the TOAR-II/ HEGIFTOM homogenized datasets

Roeland Van Malderen[1], Zhou Zang[2], Kai-Lan Chang[3,4], Robin Björklund[5], Owen R. Cooper[4], Jane Liu[2], Eliane Maillard Barras[6], Corinne Vigouroux[5], Irina Petropavlovskikh[3,7], Thierry Leblanc[8], Valérie Thouret[9], Pawel Wolff[10], Peter Effertz[3,7], Audrey Gaudel[3,4], David W. Tarasick[11], Herman G.J. Smit[12], Anne M. Thompson[13,14], Ryan M. Stauffer[13], Debra E. Kollonige[13,15], Deniz Poyraz[1], Gérard Ancellet[16], Marie-Renée De Backer[17], Matthias M. Frey[18], James W. Hannigan[19], José L. Hernandez[20], Bryan J. Johnson[7], Nicholas Jones[21], Rigel Kivi[22], Emmanuel Mahieu[23], Isamu Morino[24], Glen McConville[7], Katrin Müller[25], Isao Murata[26], Justus Notholt[27], Ankie Piters[28], Maxime Prignon[29], Richard Querel[30], Vincenzo Rizi[31], Dan Smale[30], Wolfgang Steinbrecht[32], Kimberly Strong[33], Ralf Sussmann[34]

[1]Royal Meteorological Institute of Belgium, Solar-Terrestrial Centre of Excellence, Uccle, Belgium
[2]Department of Geography and Planning, University of Toronto, Toronto, Canada
[3]Cooperative Institute for Research in Environmental Sciences, University of Colorado, Boulder, CO, USA
[4]NOAA Chemical Sciences Laboratory, Boulder, CO, USA
[5]Royal Belgian Institute for Space Aeronomy, Uccle, Belgium
[6]Federal Office of Meteorology and Climatology MeteoSwiss, Payerne, Switzerland
[7]NOAA Global Monitoring Laboratory, Boulder, CO, USA
[8]Jet Propulsion Laboratory, California Institute of Technology, Wrightwood, California, USA
[9]Laboratoire d'Aérologie, Université Toulouse III – Paul Sabatier, CNRS, Toulouse, France
[10]Observatoire Midi-Pyrénées, Université Toulouse III – Paul Sabatier, CNRS, Toulouse, France
[11]Environment and Climate Change Canada, Downsview, ONT Canada
[12]Institute of Climate and Energy Systems 3: Troposphere (ICE-3), Forschungszentrum Juelich (FZJ), Juelich, Germany
[13]Atmospheric Chemistry and Dynamics Laboratory, NASA Goddard Space Flight Center, Greenbelt, MD, USA
[14]GESTAR, University of Maryland, Baltimore County, Baltimore, MD, USA
[15]Science Systems and Applications, Inc, Lanham, MD, USA
[16]LATMOS, Sorbonne Université, Université Versailles St-Quentin, CNRS/INSU, Paris, France
[17]Groupe de Spectrométrie Moléculaire et Atmosphérique, Université de Reims, France
[18]IMKASF, Karlsruhe Institute of Technology (KIT), Eggenstein-Leopoldshafen, Germany
[19]Atmospheric Chemistry, Observations & Modeling, National Center for Atmospheric Research, Boulder, CO, USA
[20]Spanish Meteorological Agency (AEMET), Madrid, Spain
[21]School of Physics, University of Wollongong, Australia
[22]Finnish Meteorological Institute, Space and Earth Observation Centre, Sodankylä, Finland
[23]Institut d'Astrophysique et de Géophysique, Université de Liège, Liège, Belgium
[24]Earth System Division, National Institute for Environmental Studies, Tsukuba, Japan
[25]Alfred Wegener Institute, Helmholtz-Centre for Polar and Marine Research, Potsdam, Germany
[26]Graduate School of Environmental Studies, Tohoku University, Sendai, Japan
[27]Institute of Environmental Physics, University of Bremen, Bremen, Germany
[28]Royal Netherlands Meteorological Institute (KNMI), De Bilt, the Netherlands
[29]Space Earth and Environment, Chalmers University of Technology, Gothenburg, Sweden
[30]National Institute of Water & Atmospheric Research Ltd (NIWA), Lauder, New Zealand
[31]CETEMPS Dipartimento di Scienze Fisiche e Chimiche, Università degli Studi dell'Aquila, L'Aquila, Italy
[32]Deutscher Wetterdienst, Hohenpeissenberg, Germany



[33]Department of Physics, University of Toronto, Toronto, ON, Canada

[34]Karlsruhe Institute of Technology (KIT), IMK-IFU, Garmisch-Partenkirchen, Germany

*Correspondence to*: Roeland Van Malderen (roeland.vanmalderen@meteo.be)

**Abstract.** The quantification of long-term free-tropospheric ozone trends is essential for understanding the impact of human activities and climate change on atmospheric chemistry, but is challenged by the diversity between satellite tropospheric ozone

records and the sparse temporal and spatial sampling of ground-based measurements. Here, we explore if a more consistent understanding of the geographical distribution of tropospheric ozone column (TrOC) trends can be obtained by focusing on regional trends calculated from ground-based measurements. Regions were determined with a correlation analysis between modelled TrOCs at the site locations. For those regions, TrOC trends were estimated with Quantile Regression and Dynamical Linear Modelling for the Trajectory-mapped Ozonesonde dataset for the Stratosphere and Troposphere (TOST), and with a

linear mixed-effects modelling (LMM) approach to calculate synthesized trends from the homogenized HEGIFTOM (Harmonization and Evaluation of Ground-based Instruments for Free-Tropospheric Ozone Measurements) individual site trends. For different periods (1990-2021/22, 1995-2021/22, 2000-2021/22), both approaches give increasing (partial) tropospheric ozone column amounts over almost all Asian regions (median confidence), and negative trends over the Arctic regions (very high confidence). Trends over Europe and North America are mostly weakly positive (LMM method) or negative

(TOST). For both approaches, the 2000-2021/22 trends decreased in magnitude compared to the 1995-2021/22 for most of the regions, and for all time periods and regions, the pre-COVID trends are larger than the post-COVID trends. Our results enable the validation of global satellite TrOC trends, and assessment of the performance of atmospheric chemistry models to represent the distribution and variation of TrOC.

# 1 Introduction

Tropospheric ozone is a greenhouse gas and pollutant detrimental to human health and crop and ecosystem productivity (Monks et al., 2015; Fleming et al., 2018; Gaudel et al., 2018; Mills et al., 2018; Archibald et al., 2020). It is a secondary pollutant, formed as a photochemical product of oxidation reactions involving volatile organic compounds (VOCs), carbon monoxide (CO) and methane ($CH_4$) in the presence of oxides of nitrogen ($NO_x$). Stratospheric ozone influx is also a source of tropospheric ozone. Ozone losses in the troposphere occur by surface deposition and additional photochemical reactions.

Because ozone is not directly emitted, areas of ozone formation and enhanced concentrations are often geographically separated from emission sources (Fiore et al., 2009; Zhang et al., 2016; Bowman et al., 2022). The lifetime of ozone in the troposphere varies considerably with location and season, ranging from a few hours in the polluted urban boundary layer, and up to a few weeks in the free troposphere (Monks et al., 2015), with an estimated global mean tropospheric lifetime of $25.5 \pm 2.2$ days (Griffiths et al., 2021), which is long enough to be transported over hemispheric scales. Typical ozone correlation

lengths in the troposphere are about 500 km (Liu et al., 2009).





Quantification and attribution of long-term tropospheric ozone trends are critical for understanding the impact of human activity and climate change on atmospheric chemistry, but are also challenged by the limited coverage of long-term ozone observations in the free troposphere where ozone has higher production efficiency and radiative potential compared to that at the surface (Wang et al., 2022). The Tropospheric Ozone Assessment Report (TOAR) collected and analyzed long-term

tropospheric ozone observations to assess tropospheric ozone's global distribution and trends from the surface to the tropopause (Chang et al., 2017, Schultz et al., 2017, Gaudel et al., 2018, Tarasick et al. 2019). These observations include continuous measurements at remote ground-level sites (e.g. Cooper et al., 2020b), from approximately weekly ozonesondes (e.g. Thompson et al., 2021), from routine commercial aircraft (e.g. Gaudel et al., 2020), from Fourier Transform InfraRed (FTIR) spectrometers (Vigouroux et al., 2015) and retrieved by satellite instruments (e.g. Ziemke et al., 2019). With regard to

trends since the 1990s, TOAR concluded in 2018 that available observations were insufficient for the detection of an unambiguous trend in the global tropospheric ozone burden over the past two decades; in particular, the available satellite products disagreed on the sign of the trend since 2008 (Gaudel et al., 2018). However, based on studies that appeared shortly after the conclusion of the first phase of TOAR, the Intergovernmental Panel on Climate Change Sixth Assessment Report (IPCC; AR6; Sect. 2.2.5.3, Gulev et al., 2021) concluded: since the mid-1990s, free tropospheric ozone has increased by 2%–

7% per decade in most regions of the northern mid-latitudes, and 2%–12% per decade in the sampled regions of the northern and southern tropics (high confidence), and observations of tropospheric column ozone indicate increases of less than 5% per decade at southern mid-latitudes (medium confidence).

Atmospheric chemistry models have been extensively used for quantifying the drivers of ozone trends and for estimating ozone radiative impacts, and show that global-scale increases (largest at northern mid-latitudes) in tropospheric ozone since pre-

industrial times are driven by anthropogenic emissions of ozone precursor gases (Young et al., 2018; Archibald et al., 2020; Griffiths et al., 2021; Christiansen et al., 2022, Wang et al., 2022; Fiore et al., 2022). However, their applications to the continental and global scales are largely constrained by the limited coverage of robust long-term ozone measurements for evaluating modelled ozone trends, especially in the free troposphere (Wang et al., 2022).

One of the scientific scopes of the second phase of TOAR, TOAR-II, is "to provide an observation-based, up-to-date

assessment of tropospheric ozone's distribution and trends on regional, hemispheric and global scales. Observations include in situ measurements using modern quantitative methods (e.g. UV-absorption instruments - surface and airborne), wet chemical ozonesondes, and remote sensing methods from ground-based and space-based platforms (e.g. lidar, UV-absorption, thermal-infrared)" (see Cooper et al., 2020a). One TOAR-II Focus Working Group, HEGIFTOM (Harmonisation and Evaluation of Ground-based Instruments for free-tropospheric ozone measurements, https://hegiftom.meteo.be) had a major objective to

provide quality assessed tropospheric ozone data sets, with uncertainties. In Van Malderen et al. (2024), the tropospheric ozone distribution and trends from these homogenized ground-based and in-situ measurement sites (n=55) are shown for the period 2000-2022. One of the findings of the study is that no geographically consistent patterns emerge from the distribution of the individual site trends, except that 10 out of 11 Arctic sites (> 55°N) display negative tropospheric ozone column trends. Sites with shorter time series or with time series with large gaps could not be included in this study, by lack of a reliable trend





estimate. On the other hand, the included sites have a large diversity in monthly sampling frequencies (ranging between 2 and 25 observations per month on average), whereas several studies (Logan, 1999; Chang et al., 2020, 2022, 2024) suggest that around 15 observations per month are required to calculate tropospheric ozone trends with high confidence. To cope with ground-based or in-situ short time series and time series with gaps, and to increase the monthly sampling frequency, tropospheric ozone measurements have been merged or fused in a large number of studies for (regional) trend estimation (e.g.

Cooper et al., 2010; Chang et al., 2020, 2022; Gaudel et al., 2020, 2024; Steinbrecht et al., 2021; Thompson et al., 2021; Wang et al., 2022). Here, we follow two different approaches to calculate regional trends based on combining ground-based or in-situ measurements. The first approach is based on the Trajectory-mapped Ozonesonde dataset for the Stratosphere and Troposphere (TOST, see Zang et al., 2024), 3-dimensional global-scale ozone dataset using a trajectory-mapping method, extending sparse ozonesonde measurements and filling gaps in the spatial domain by backward and forward trajectory

simulation. This dataset has been included in the assessment of (the latitudinal variation of) tropospheric ozone column trends in the latest IPCC report (Gulev et al., 2021). The second approach calculates regional trends from the 5 different measurement techniques involved in HEGIFTOM by synthesizing the trends from the individual time series using linear mixed-effects modelling.

      To determine the extension of the regions for which we want to calculate trends, guidance is provided by a correlation analysis

between tropospheric ozone columns from an atmospheric chemistry reanalysis model at the ground-based site locations.

      This paper is organized as follows. First, we describe the data and model output used in Section 2. In Section 3, the principles of the correlation analysis and the model for calculating synthesized trends is presented. Section 4 shows the (partial) tropospheric ozone column distribution derived from TOST. Regional trend estimates of (partial) tropospheric ozone columns for both approaches are depicted and compared in Section 5. In the final section 6, we draw the conclusions and provide an

outlook for further work.

## 2 Data

### 2.1 Homogenized ground-based and in-situ observations

      In Van Malderen et al. (2024), the available ground-based and in-situ measurements that are used here are presented in great detail. Basically, the tropospheric ozone data from 5 different measurement platforms are used to calculate synthesized trends

in this study: ozonesondes, In-service Aircraft for a Global Observing System (IAGOS), Brewer/Dobson Umkehr, FTIR, and Lidar. Within the HEGIFTOM (Harmonization and Evaluation of Ground-based Instruments for Free-Tropospheric Ozone Measurements) Focus Working Group in the second phase of the Tropospheric Ozone Assessment Report (TOAR-II), harmonization efforts within each of those networks have been conducted, and uncertainty estimates for the measurements are provided. These activities have been described in Van Malderen et al. (2024) and on the HEGIFTOM website

(http://hegiftom.meteo.be). It is important to mention here that only sites that participated in this homogenization activity are used in this analysis, at the cost of losing observations in already very poorly sampled regions (e.g. by not including the non-



homogenized Japanese and Australian ozonesonde records). Non-homogenized time series might include changes in the mean (breakpoints) in their time series due to instrument or methodological changes, and therefore cannot be used within the context of trend estimation.

The ozonesondes, IAGOS, and Lidar techniques provide (tropospheric) ozone profile measurements with high vertical resolution, while the Brewer/Dobson and FTIR retrievals have basically around 1 degree of freedom in the troposphere, resulting in a tropospheric ozone column measurement. This is the tropospheric ozone column (TrOC) metric used here, which extends from the surface up to about 300 hPa. It should be noted that the upper limit of the tropospheric column is not well-defined in these cases. The ozone mixing ratio profile measurements of ozonesondes, IAGOS and Lidar are integrated to

calculate the entire tropospheric ozone column TrOC metric (surface to 300 hPa), which is column-averaged by dividing by the extent (in pressure) of the tropospheric column. The lower pressure limit of 300 hPa is chosen because this is more or less the global upper limit (cruising altitude) of the IAGOS aircraft. In addition, for ozonesondes, IAGOS, and Lidar, a free-tropospheric column-averaged ozone column metric (FTOC) is defined here as the integrated ozone mixing ratios between 700 and 300 hPa. For all techniques, these partial tropospheric ozone columns are available through the HEGIFTOM website

(https://hegiftom.meteo.be/datasets/tropospheric-ozone-columns-trocs), together with their uncertainty estimates. In this paper, the synthetized regional trends are calculated from all the available measurements (the so-called L1-version), and no time averaging (to daily, L2 version, or monthly means, L3) is applied.

## 2.2 TOST

The Trajectory-mapped Ozonesonde dataset for the Stratosphere and Troposphere (TOST) provides a global-scale and long-

term (1970-2021) ozone climatology (Liu et al., 2013a, b; Zang et al., 2024). TOST is generated in 3-dimensional grids of 5º×5º×1 km by latitude, longitude, and altitude at 26 1-km layers from the surface to 27 km or at 26 pressure levels from the surface to 20 hPa (Zang et al., 2024). TOST depends on neither a prior nor photochemical modelling and thus provides insights independent from satellite datasets and model simulations. TOST is generated based on trajectory mapping, which extends the sparse ozonesonde measurements using 4-day forward and backward trajectories starting from 26 1-km levels of each

ozonesonde profile. The same 26 trajectories at 1 km interval are simulated using the Hybrid Single Particle Lagrangian Integrated Trajectory (HYSPLIT) model (Draxler and Hess, 1998) driven by the reanalysis of meteorological data from the National Centers for Environmental Prediction/National Center for Atmospheric Research (NCEP/NCAR; Kalney et al., 1996). Rather than simple linear or polynomial interpolation, the ozone measurement from the ozonesonde profile at an origin of a trajectory is assigned along its forward and backward trajectory paths. Finally, ozone mixing ratios in each grid are averaged

over all trajectories passing that grid in the period of interest, usually in a month. Meticulous validations show that TOST agrees well with ozonesonde (relative difference of 2-4%) and aircraft (relative difference of 5%) measurements in the troposphere (Zang et al., 2024).

Although trajectory mapping has extended the spatial domain of ozonesonde measurements, large gaps still remain where ozonesondes are sparse. Consequently, there are missing data at one or more levels when integrating ozone mixing ratio over





certain layers to derive TrOC. Two gap-filling strategies are applied. First, if there are >80% available data in a column from the surface to 300 hPa (or 700 to 300 hPa) and the gaps are not consecutive, the gaps are filled by interpolation vertically. Second, gaps at each layer are interpolated horizontally using a distance-sample-weighted-average method (Liu et al. 2022a), which fills gaps by the mean ozone concentrations of the surrounding grid cells with valid data, weighted by their distances and sample sizes. Finally, to ensure our results are robust with sufficient data samples, we derive TrOC based on the annual

mean ozone value, which requires at least one available data in each season.

**2.3 CAMS**

For a spatial representativeness analysis to define regions that are spatially cohesive in tropospheric ozone column amounts, we use the CAMS (Copernicus Atmosphere Monitoring Service, Peuch et al., 2022) reanalysis data. This reanalysis supplies ozone data every 3 hours from 2003 to 2023 and combines model data and observations to give gridded monthly mean time

series (0.75°x0.75°) on 25 different pressure levels; the data were downloaded from https://ads.atmosphere.copernicus.eu/datasets/cams-global-reanalysis-eac4?tab=overview. First, we calculated ozone column densities in DU from the CAMS mass-mixing ratios for each vertical layer using $N_A/M_{air}/g*r_m*\Delta p/N_{conv}$ (where $N_A$ is Avogadro's constant, $M_{air}$ is molecular mass of air, g is the gravitational acceleration, $r_m$ is mass mixing ratio of ozone, $\Delta p$ is pressure difference between layer boundaries, and $N_{conv}$ is a conversion number 2.6867e20 molec/cm$^2$). To obtain the

tropospheric ozone column TrOC (between 1000 hPa and 300 hPa) and the free-tropospheric ozone column between 700 hPa and 300 hPa (FTOC), we add the column densities in DU of the pressure levels within these partial column layers. Finally, the gridded monthly averaged time series are deseasonalized into relative anomalies by subtracting and dividing each column value by the long-term mean value of its corresponding month in the year.

**3 Methods**

**3.1 Definition of regions with correlation analysis**

In the literature, (vertical profile) ozone measurements at different locations have been merged or fused into regions to study the present-day tropospheric ozone distribution and trends at a larger spatial scale (e.g. Cooper et al., 2010; Chang et al., 2020, 2022; Gaudel et al., 2020, 2024; Steinbrecht et al., 2021; Thompson et al., 2021; Wang et al., 2022). In those studies, the sampling regions were determined based on the distance of the different measurement locations, in addition to the availability of frequent sampling in both the early and late periods of the considered time range (Gaudel et al., 2020). In this paper, we

define the regions for merging individual sites not only based on the distance between sites, but also driven by the expected spatial correlation in tropospheric ozone column between the sites. A major difference between our study and the aforementioned studies is that we are considering tropospheric ozone columns, instead of tropospheric ozone profiles. We use the regions defined in Gaudel et al. (2020) and Wang et al. (2022) as the starting points for merging time series, but the final

regional domains are refined based on the spatial correlation characteristics.





To assess the spatial representativeness of the HEGIFTOM stations before merging them, we follow the methodology of Weatherhead et al. (2017), who investigate how well an individual station represents changes occurring at other nearby locations. The authors argue that this can best be addressed with long-term observations at many nearby sites, which is often not possible due to sparse data availability. Therefore, they propose that this question can be addressed with satellite data, as

in their paper, or with high-quality model output. The Pearson correlation coefficient r between time series of deseasonalized monthly averages is chosen in their study to serve as an appropriate metric to describe representative observations for understanding global representativeness. Correlations of r > 0.7 are referred to as "well correlated" and r > 0.9 and above as "strongly correlated". The same approach was followed in an earlier study by Sofen et al. (2016), in which the spatial representativeness of the global surface ozone network was assessed by making use of an Eulerian forward atmospheric

chemical transport model (GEOS-Chem). Translated to our study here, we use the CAMS high-quality reanalysis gridded output of deseasonalized monthly averages of tropospheric ozone column amounts (see Sect. 2.3). At each HEGIFTOM site location grid cell, the Pearson correlation coefficient is calculated between its TrOC and the TrOCs at any other grid point within the CAMS grid. This is done both for the surface to 300 hPa and 700-300 hPa tropospheric ozone column metrics. This processing results in correlation maps for each HEGIFTOM site location, of which two examples are shown in Fig 1. Those

correlation maps are then used to determine which sites to combine, by checking the correlation coefficients of a station with every other station in the same region and applying the r > 0.7 threshold value ("well correlated"). Although the free-tropospheric ozone column metric resulted in slightly higher correlation coefficients between the grid points than the surface to 300 hPa metric, both metrics produced identical regions. As can be seen in Fig 1a, all of continental Europe seems to be well-correlated in terms of tropospheric ozone column monthly anomalies, while a further subdivision of East Asia, as defined

in Wang et al. (2022), is needed to define well-correlated regions (see Fig. 1b). In total this method produced 24 well-correlated regions, e.g. Continental Europe, the European Arctic, Canadian Arctic, California, Western North America (Western USA and Pacific Northwest), Eastern North America, Southeast US, East Asia (East China, Northeast Asia, South Japan), Southeast Asia (Southeast Asia, South China Sea), Malaysia/Indonesia (Indonesia, Southern Malay Peninsula), India, Persian Gulf (Persian Gulf, East Mediterranean Sea), Gulf of Guinea (Gulf of Guinea, West African Highlands), Northern South America

(Middle America, Caribbean), Hawaii, and Oceania. This list of regions forms the starting point for the regional trend estimations, for both TOST and the synthetized trend approaches, but the spatial and temporal sampling of the ground-based observations in those regions will put further constraints on the final determination of the regions.

A last remark concerns the use of the CAMS reanalysis ozone output to define well-correlated regions instead of e.g. the HEGIFTOM observations themselves. As some of the HEGIFTOM time series have limited spatial and temporal sampling,

the calculation of correlation coefficients directly between those time series can become problematic. This is exactly the argument used by Weatherhead et al. (2017) for choosing satellite data or high-quality model output for this purpose. The authors use the same reasoning to explain that the agreement between such satellite or high-quality model output and individual-site data is challenged by spatial averaging issues, temporal sampling issues, as well as fundamental measurement issues. In particular, we find a mean correlation between CAMS and HEGIFTOM TrOC amounts of about 0.75 for monthly



averaged data, but around 0.9 for IAGOS Frankfurt, and European ozonesonde stations (Uccle, Payerne, Hohenpeissenberg, Madrid, Legionowo). However, the correlation coefficients drop to a mean value of around 0.22 for the deseasonalized monthly averages. This shows that the CAMS model ozone output does not fully mimic the HEGIFTOM observations (or the other way around). But, as the CAMS model is only used here to define the boundary limits of the regions for which synthetized trends from combined observations are calculated, this does not pose a problem here.


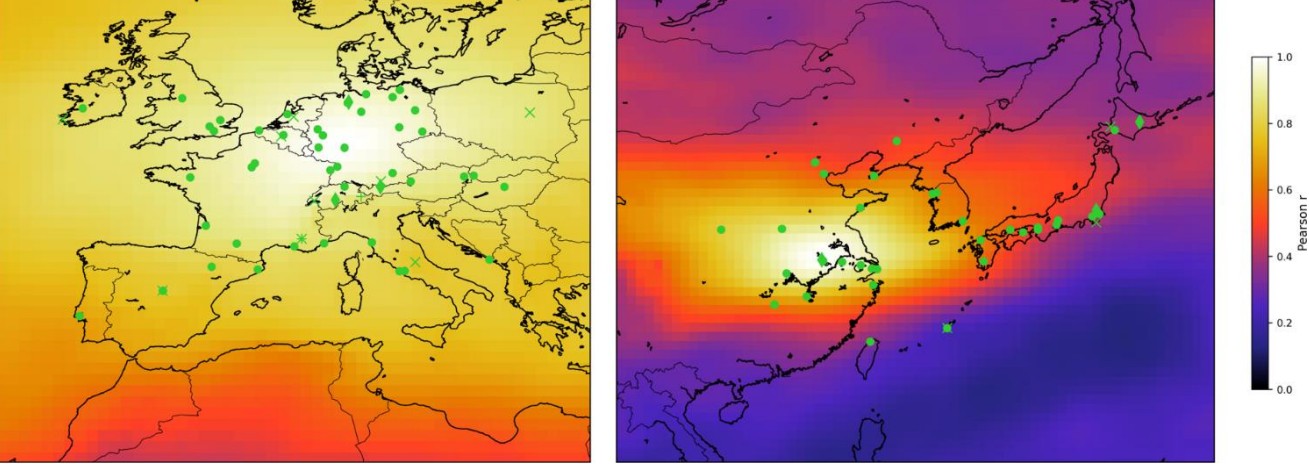

**Figure 1: (a) Correlation map at the IAGOS FRA airport location, constructed by calculating the linear Pearson correlation coefficients between the CAMS TrOC monthly anomaly grid point at the IAGOS FRA airport location and the CAMS TrOC monthly anomaly grid points in the European domain. (b) Same as (a), but now for the FTOC at the FTIR Hefei location in East**
**China. The different HEGIFTOM sites are marked with green symbols, with dots for IAGOS, × for ozonesondes, diamonds for FTIR, plus for Umkehr, squares for Lidar.**

**3.2 Linear mixed-effects modeling for deriving synthesized trends**

Time series measured at nearby locations typically yield a stronger correlation than those far away. To gain more insight on a broader perspective, it is natural to pursue more effective evidence by assembling sites with an acceptable correlation, instead
of dealing with each record separately. The impetus for studying trends from multiple correlated time series is to not only add additional value through evidence synthesis (Richardson, 2022), but also provide more clear implications for a larger scale variability (Chang et al., 2022). The rationale of synthesized trends can be viewed as a two-level modeling in statistics, one needs to consider both the trends at individual sites (bottom level, also known as the random effect) and the overall trends representing the baseline for all sites (top level, also known as the fixed effect). Statistically, the class of linear mixed-effects
models (LMM) is a standard method to deal with multilevel or hierarchical problems (Gelman and Hill, 2006). Instead of averaging various data sources in advance, the LMM approach aims to include all the data sources into the regression model, and account for potential differences between data sources based on regression specifications. Let y(t,k) be the time series observed at time t from k-th data platforms, the statistical model can be expressed as:



$$y(t,k) = a + bt + \alpha_k + \beta_k t + c_1 \sin\left(Month \times \frac{\pi}{6}\right) + c_2 \cos\left(Month \times \frac{\pi}{6}\right) + \tag{1}$$

$\quad c_3 \sin\left(Month \times \frac{\pi}{3}\right) + c_4 \cos\left(Month \times \frac{\pi}{3}\right) + N(t,k),$

where $a + bt$ represents the synthesized trends (the fixed effect); $\alpha_k$ and $\beta_k$ are the site-specific adjustments (the random effect) to the synthesized trends, so for each site the trends are $(a + \alpha_k) + (b + \beta_k)t$; 4 harmonic terms are used to account for the seasonality, and $N(t,k)$ is the residual term. Across different data sources, the expected values $E(\alpha_k) = 0$ and $E(\beta_k) = 0$, indicating that individual sites may reveal differences, but the overall tendency can be represented by the synthesized trends.

It should be noted that our LMM uses an intercept and a slope to adjust the difference from each individual source against the overall trends. If various data sources have overlapping time series in the study period (i.e. repeated measurements are the key to understanding aleatoric and epistemic uncertainties), more detailed nonlinear random effects can be investigated, such as nonlinear trend differences or seasonal discrepancy at each individual site.

LMM does not require that trends at individual sites follow the same direction/conclusion, but assume the synthesized trends

to be representative based on available sampled sites. Therefore, we assemble sites and regions that are well-correlated spatially, as selected from the correlation analysis described here in Sect. 3.1. A small example of this technique for a subset of western European sites (2 ozonesondes, 3 IAGOS airports) is given in Fig. 2. From this plot, it can be seen that LMM estimates the synthesized trend a+bt from the monthly mean time series collected at the different sites or airports. But, to adjust the differences from each dataset, LMM also estimates the coefficients of adjustment for each source (instead of directly

calculating trends for each site). Importantly, it should be noted that the average of all adjustment coefficients (5 for intercept and 5 for slope) yield zero, which indicates that an average of all site trends represents the overall trend.

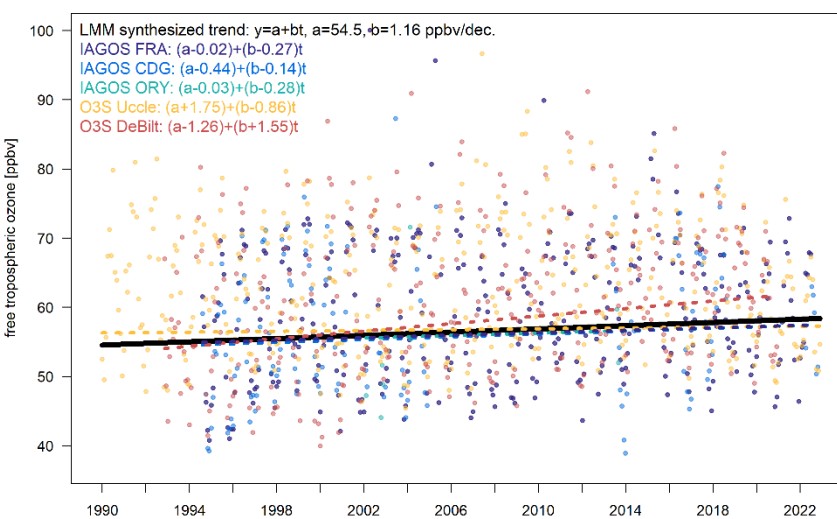

**Figure 2: Illustration of the linear mixed-effects modelling method for the monthly mean time series of IAGOS FRA (Frankfurt),**
**CDG, ORY (both Paris), Uccle and De Bilt ozonesondes. The LMM synthesized trend is shown in full black, and the coloured dashed**
**lines denote the trend lines of the individual time series contributing to the LMM trend. In the legend, it can be seen how those**
**individual trends have been adjusted to calculate a synthesized trend for the combined time series.**





In practice, the LMM method will be applied on the L1 (all measurements) data. As a consequence, sites with more measurements will naturally weigh more on the synthesized trend estimation. However, we additionally include the mean uncertainties associated with the different techniques as weights in the LMM calculations. By simply averaging the TrOC

uncertainties over the different sites by technique, the following values are obtained: 2.5% for lidar, 5.5% for ozonesonde and IAGOS, 14% for FTIR, and 15% for Umkehr. Sites with a small number of measurements, either in terms of sampling frequency within a month or a rather limited time series due to gaps or the absence of a long-term monitoring program, are expected to contribute only marginally to the synthesized trend. To assess this, we undertook a sensitivity analysis and calculated LMM trends including all sites, and including sites with at least 30, 60, and 120 months of data availability. The

findings are described in Appendix A. It follows that applying the LMM trend estimation for sites that have at least 30 monthly values available is a good compromise between limiting the impact of trends at individual sites with a very limited amount of data and retaining a useful number of regions. Therefore, in the remainder of the paper, we will show trend estimates from this LMM variant only, and the sites retained in each region are listed in Tables 1 and S1, and shown in Fig. 3. Taking the data coverage criteria developed in Gaudel et al. (2024) into account, based on the percentage of months with observations available

and the average number of observations per month, the regions with high data coverage (see Table 2) are Europe, both Arctic regions, Western USA, Hawaii and Oceania. The regions that can be classified as moderate data coverage are California, the Pacific Northwest, Eastern North America, Southeast USA, East China, and Northeast Asia. All the other regions have low data coverage.

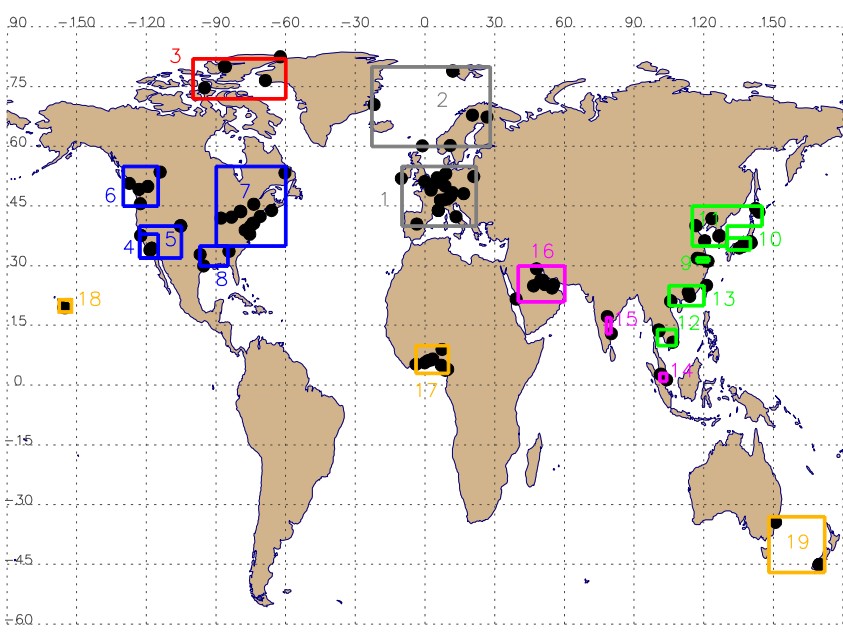

**Figure 3: Map showing the 19 different regions and the individual HEGIFTOM ground-based sites (see Tables 1 and S1) that are used for the regional synthetized trends calculation using the linear mixed-effects modelling (LMM) approach.**





LMM estimations of the synthesized trends are implemented through the gamm (generalized additive mixed model) environment, built in the R package mgcv (Wood, 2017). It is desirable to ensure that our aggregate sites are a better regional

representation, instead of merely boosting sample size for a lower uncertainty. We use a block bootstrap approach to estimate the synthesized trend uncertainty (Lahiri, 2013), which is less dependent on the sample size than conventional standard error. The code for implementing the synthesized trend estimation is provided in the supplementary material, section S1.

**Table 1: Overview of the different regions identified by the CAMS TrOC/FTOC monthly anomaly correlation analysis and finally**
**used for the LMM synthesized trends estimation. For each region, the geographical limits and the contribution HEGIFTOM instruments and sites are listed. The numbers before the region are associated with the numbers marked on Fig. 3. More details on the individual sites or airports are provided in Table S1.**

| Region | Geographical extent | Instrument | Sites/Airports |
|---|---|---|---|
| **1. Europe** | 10°W – 20°E, 40 – 50°N | IAGOS | VIE, MUC, CDG, FRA, BRU, LGW, DUS |
| | | O3S | Madrid, l'Aquila, OHP, Payerne, Hohenpeissenberg, Uccle, Valentia, De Bilt, Legionowo |
| | | Umkehr | 40 (OHP), 35 (Arosa) |
| | | FTIR | Jungfraujoch, Zugspitze, Bremen |
| | | Lidar | OHP |
| **2. European Arctic** | 20°W – 25°E, 60° – 80°N | O3S | Lerwick, Sodankylä, Scoresbysund, Ny Ålesund |
| | | FTIR | Harestua, Kiruna, Ny Ålesund |
| **3. Canadian Arctic** | 100° – 60°W, 75° – 80°N | O3S | Resolute, Eureka, Alert |
| | | FTIR | Thule, Eureka |
| **4. California** | 123° – 115°W, 32° – 38°N | IAGOS | LAX, SFO |
| | | Lidar | TMF |
| **5. Western United States** | 123° – 105°W, 32° – 40°N | IAGOS | LAX, SFO |
| | | O3S | Boulder |
| | | Umkehr | 67 (Boulder) |
| | | FTIR | Boulder |
| | | Lidar | TMF |
| **6. Pacific Northwest** | 130° – 115°W, 45° – 55°N | IAGOS | PDX, YVR |
| | | O3S | Kelowna, Edmonton, Port Hardy |
| **7. Eastern North America** | 90° – 60°W, 40° – 55°N | IAGOS | IAD, PHL, JFK, EWR, ORD, DTW, BOS, YYZ, YUL |
| | | O3S | Wallops Island, Yarmouth, Goose Bay |
| | | FTIR | Toronto |
| **8. Southeast US** | 97° – 85°W, 30° – 35°N | IAGOS | IAH, DAL, ATL |





| 9. East China | 117° – 122°E, 31°N | IAGOS | PVG, SHA, NKG |
|---|---|---|---|
| | | FTIR | Hefei |
| 10. South Japan | 130° – 140°E, 35°N | IAGOS | KIX, NGO, NKM, NRT |
| | | FTIR | Tsukuba |
| 11. Northeast Asia | 115° – 145°E, 35° – 45°N | IAGOS | TAO, ICN, GMP, PEK, SHE |
| | | FTIR | Rikubetsu, Moshiri |
| 12. SE Asia | 100° – 107°E, 10° – 15°N | IAGOS | SGN, BKK, DMK |
| 13. South China Sea | 105° – 120°E, 20° – 25°N | IAGOS | HAN, HKG, CAN, TPE |
| | | O3S | Hanoi |
| 14. Southern Malay Peninsula | 101° – 104°E, 1° – 3°N | IAGOS | SIN, KUL |
| | | O3S | Kula Lumpur |
| 15. India | 78° – 80°E, 13° – 17°N | IAGOS | MAA, HYD |
| 16. Persian Gulf | 40° – 60°E, 21° – 30°N | IAGOS | JED, AUH, RUH, DXB, DOH, BAH, DMM, KWI |
| 17. Gulf of Guinea | 4°W – 10°E, 3° – 9°N | IAGOS | SSG, DLA, PHC, ABJ, ACC, LFW, COO, LOS, ABV |
| 18. Hawaii | 155°W, 20°N | O3S | Hilo |
| | | Umkehr | 31 (Mauna Loa) |
| | | FTIR | Mauna Loa |
| 19. Oceania | 150° – 170°E, 35° – 45°S | O3S | Lauder |
| | | Umkehr | 256 (Lauder) |
| | | FTIR | Lauder, Wollongong |

## 3.3 Dynamical Linear Modelling on monthly mean datasets

Dynamical linear modelling (DLM) allows for the determination of a nonlinear time-varying trend from a monthly mean time series. This Bayesian approach regression fits the data time series for a nonlinear time-varying trend and seasonal and annual modes (Alsing et al., 2019). The model used allows for a variability of the sinusoidal seasonal modes and includes the autoregressive (AR1) correlation process with variance and correlation coefficient as free parameters in the regression. The estimation of the posterior uncertainty distribution is performed with the Markov chain Monte Carlo (MCMC) method and

considers the uncertainties on the seasonal cycle, on the autoregressive correlation and on the nonlinearity of the trend. DLM has been used in Van Malderen et al. (2024) and shows good agreement with QR and MLR trend estimations on tropospheric ozone partial columns. While QR and LMM estimate long-term, i.e. decadal, trends, DLM allows here to investigate the potential non-linear nature of the trends in the considered time interval and therefore possibly highlight any partial (in time) trend significance. The linear trend is allowed to change continuously at each time step. Each trend value relies on a running

one-year time range, with the information being weighted by the variance of the ozone data and by an allowed variability of the linear trend estimate. The trend variations are then given in ppbv/yr or %/yr.




**Table 2: Time range and statistical information for the different identified regions from Table 1. For each region, the start (1) and end year (2) of the combined time series are shown, (3) the total number of observations during this time period, (4) the percentage of the months that have data available in this time period, (5) the average number of observations in those months that have observations, (6) the largest data gap (in months) during the time period, (7) the percentage of months that have at least 4 (or 12, between brackets) observations available in that month, and (8) data coverage. Data coverage criteria as in Gaudel et al. (2024) are applied: high coverage if column (4) > 90% and column (5) > 15, moderate coverage if 66% < column (4) < 90% and 7 < column (5) < 15, or only one high coverage criterion fulfilled, and low coverage if column (4) < 66% and column (5) < 7.**

|  | (1) | (2) | (3) | (4) | (5) | (6) | (7) | (8) |
|---|---|---|---|---|---|---|---|---|
|  | begin | end | $N_{obs}$ | % months with obs | $N_{avg}$ in months with obs | Largest data gap (months) | % months with 4 (12) obs | coverage |
| Europe | 1990 | 2022 | 92548 | 100.0 | 233.71 | 0 | 100 (100) | high |
| Eur Arctic | 1990 | 2022 | 17554 | 94.4 | 46.94 | 0 | 94.2 (90.4) | high |
| Can Arctic | 1990 | 2021 | 16999 | 97.7 | 43.93 | 6 | 97.5 (68.4) | high |
| California | 1994 | 2022 | 3412 | 76.8 | 11.22 | 9 | 67.2 (33.8) | moderate |
| W USA | 1990 | 2022 | 10970 | 98.2 | 28.20 | 5 | 96.0 (87.9) | high |
| Pacific NW | 1990 | 2021 | 3224 | 95.0 | 8.57 | 1 | 81.8 (18.7) | moderate |
| E N America | 1994 | 2022 | 16559 | 86.1 | 48.56 | 0 | 86.1 (84.6) | moderate |
| SE US | 1994 | 2022 | 3781 | 63.1 | 15.12 | 12 | 54.0 (28.8) | moderate |
| E China | 1997 | 2021 | 3826 | 38.4 | 25.17 | 22 | 23.0 (13.1) | moderate |
| S Japan | 1994 | 2020 | 3658 | 65.2 | 14.18 | 26 | 58.8 (36.4) | low |
| NE Asia | 1994 | 2022 | 4213 | 71.2 | 14.94 | 12 | 57.1 (32.6) | moderate |
| SE Asia | 1994 | 2022 | 1259 | 28.8 | 11.04 | 68 | 20.2 (9.1) | low |
| S China Sea | 1994 | 2021 | 2705 | 58.3 | 11.71 | 27 | 23.0 (8.8) | low |
| S Malay Peninsula | 1995 | 2022 | 755 | 70.5 | 2.71 | 23 | 11.6 (1.0) | low |
| India | 1994 | **2018** | 945 | 38.9 | 6.14 | 48 | 24.5 (5.1) | low |
| Persian Gulf | 1997 | 2022 | 1763 | 39.7 | 11.23 | 67 | 34.6 (14.9) | low |
| Gulf of Guinea | 1997 | 2022 | 1450 | 38.1 | 9.60 | 37 | 28.8 (13.4) | low |
| Hawaii | 1990 | 2022 | 21937 | 99.2 | 55.82 | 3 | 92.9 (87.6) | high |
| Oceania | 1990 | 2021 | 23967 | 100.0 | 60.52 | 0 | 100 (84.1) | high |

## 4 Tropospheric regional ozone column distribution

Figure 4 shows the tropospheric ozone distributions over two altitudinal columns, TrOC being the mean mixing ratios from the surface to 300 hPa (Fig. 4a) and FTOC being the mean mixing ratios from 700-300 hPa (Fig. 4b). Both TrOC and FTOC are highest in the Arctic and the lowest in the tropics. This is mainly due to the latitudinal variation in the tropopause height that is lowest in the Arctic (~9 km) and highest in the tropics (~17 km) (Liu et al., 2013b). High ozone hotspots are also observed in the outflows of Asia and North America with TrOC reaching 60 ppb and FTOC reaching 70 ppb. Compared to TrOC, FTOC is overall higher at mid-latitudes where stratosphere-to-troposphere transport is most active. In addition, the





ozone precursor emissions are highest and thus promote ozone generation there (Zhang et al., 2016). At midlatitudes, the Middle East and northern Africa show a hotspot for TrOC (>65 ppb) and FTOC (>70 ppb), which agrees well with the TrOC distributions from satellite and modelling data (Liu et al., 2022b). The column-averaged independent samples are presented

for the corresponding TrOC (Fig. 4c) and FTOC (Fig. 4d). The independent samples for a grid cell are the number of trajectories passing through that grid cell, i.e., a trajectory is counted only once regardless of how long that trajectory remains in that cell, while the column-averaged mean takes the average over all the layers in the column. The more trajectories passing through a grid cell, the more data samples and the lower the standard error for that cell. Obviously, the regions of dense ozone soundings, such as Europe and Eastern North America, have more independent samples. Based on the independent samples, as well as

the regions determined by the correlation analysis (see Section 3.1), we selected 12 regions (Fig. 4a and Table 3) for analyzing TOST-based TrOC and FTOC trends. Within each region, high spatial correlation (>0.7, in line with Weatherhead et al., 2017) is found among stations. Table 3 lists the column-averaged mean independent samples over 1995-2021 in the 12 regions.

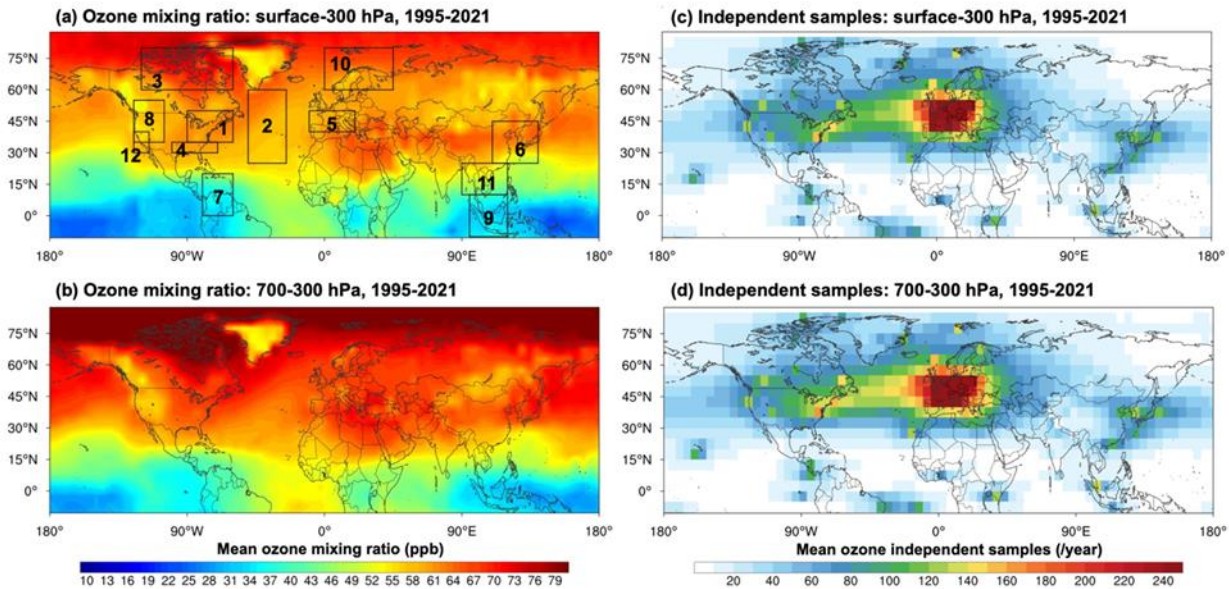

**Figure 4: (a, b) TOST-based spatial distributions of the mean ozone mixing ratios over the columns from the surface to 300 hPa (a) and of 700 to 300 hPa (b) during 1995-2021. (c-d) The same as (a-b), but for the corresponding mean independent samples. The boxes and ID numbers in (a) denote 12 regions for trend analysis (see Table 3 for the extent of each region).**





**Table 3. Information on the defined 12 regions for TOST-based trend calculations, including each region's ID number labelled in Fig. 4a, name and extent.**

| ID | Region name | Region extent | Regional mean independent samples (counts/year) | | |
|----|-------------|---------------|---------|---------|---------|
| | | | Sfc-300 | Sfc-700 | 700-300 |
| 1 | Eastern North America | 90° – 60° W, 35° – 50° N | 81 | 56 | 90 |
| 2 | Northern Atlantic Ocean | 65° – 30° W, 25° – 60° N | 57 | 39 | 65 |
| 3 | Canadian Arctic | 120° – 60° W, 60° – 80° N | 15 | 11 | 17 |
| 4 | Southern USA | 100° – 70° W, 30° – 35° N | 54 | 41 | 59 |
| 5 | Continental Europe | 10° W – 20° E, 40° – 50° N | 202 | 176 | 218 |
| 6 | East Asia | 110° – 140° E, 25° – 45° N | 55 | 46 | 59 |
| 7 | Northern South America | 80° – 60° W, 0° – 20° N | 19 | 17 | 21 |
| 8 | Western North America | 125° – 105° W, 35° – 55° N | 53 | 31 | 56 |
| 9 | Malaysia/Indonesia | 95° – 120° E, 10° S – 10° N | 29 | 21 | 32 |
| 10 | European Arctic | 0° – 45° E, 60° – 80° N | 24 | 20 | 26 |
| 11 | Southeast Asia | 90° – 120° E, 10° – 25° N | 29 | 25 | 31 |
| 12 | California | 125° – 115° W, 30° – 40° N | 42 | 23 | 47 |

## 5 Trends

### 5.1 TOST regional trends

In this section, we present the TOST trends for TrOC and FTOC calculated by quantile regression (QR), which is considered to be a robust method for trend analysis when there are many outliers and intermittent missing values in a time series (Davino et al., 2013). When calculating the QR trends, it is required that there is >80% data availability in the time series to ensure robust results. To better understand the differences between TrOC and FTOC trends, we also calculated the QR trends for the lower-troposphere column-averaged ozone mixing ratio from the surface to 700 hPa, LTOC. Moreover, in an effort to estimate if TrOC or FTOC trends may be non-linear or partly significantly different from zero, DLM (see section 3.3) trends estimates have been computed on the TOST regional datasets.

### 5.1.1 Spatial distributions of TrOC trends

Figure 5 shows the spatial distributions of TrOC and FTOC trends over 1995-2021. Because of the different magnitudes in TrOC and FTOC mixing ratios, both absolute and relative trends are shown for justified comparison between TrOC and FTOC. Both TrOC and FTOC show significant increases in East Asia and its outflow, from 90°E to 150°W. The absolute trends of





TrOC and FTOC in East Asia appear similar, yet the relative TrOC trend is slightly larger than that for FTOC (Fig. 5d and 5e),
due to the fact that LTOC increases faster than FTOC there (Fig. 5c). Spatially, TrOC, FTOC, and LTOC share similar variation
patterns. In contrast, the trends of TrOC outflows from North America appear decreasing. FTOC over the Middle East and
northern Africa FTOC shows increasing trend, being significant at some places. This indicates that the strong increase in LTOC
is likely the main reason for increasing trends of TrOC over the Middle East and northern Africa. For North America and
Europe, generally weak decreasing trends are found in TrOC and FTOC. However, LTOC shows decreasing trends in North
America, but generally weak increasing trends in Europe.

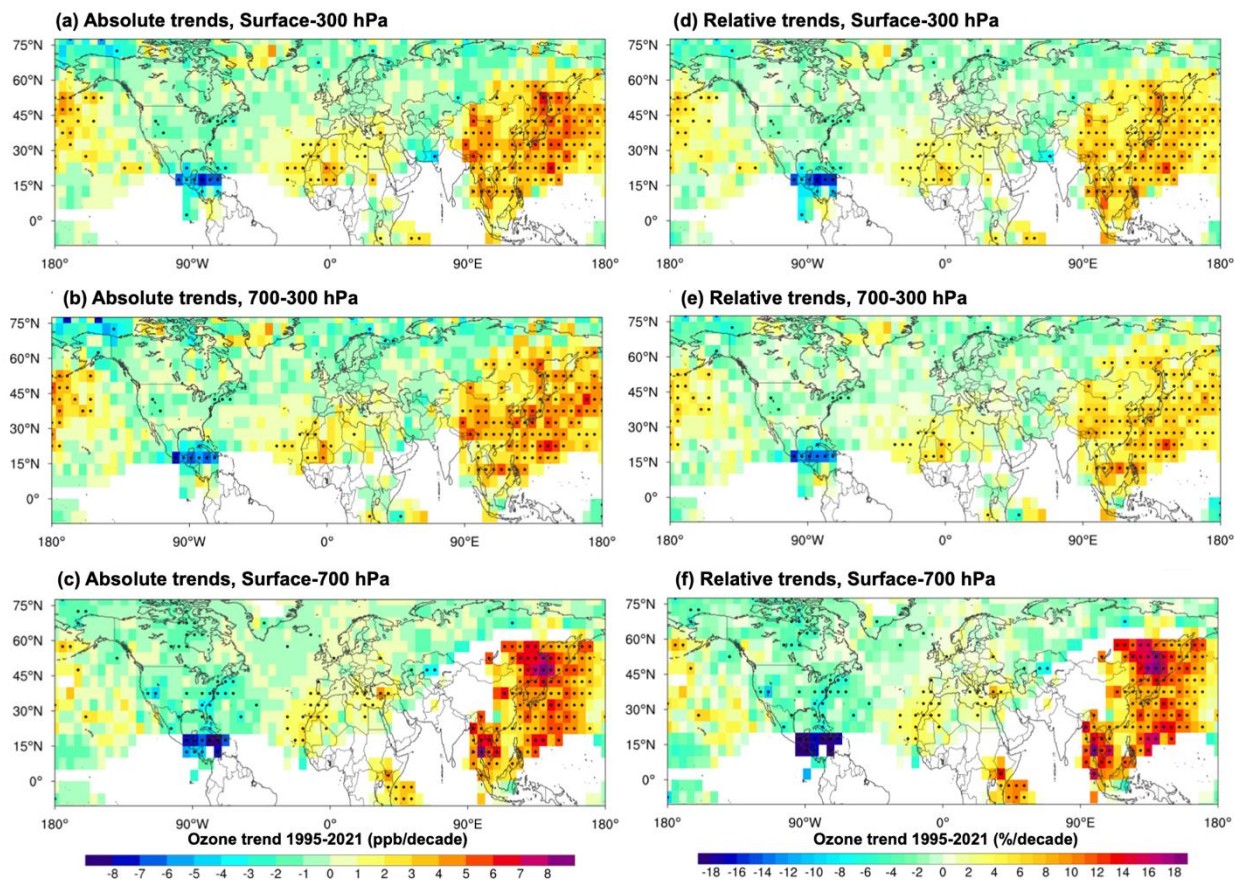

**Figure 5: (a-c) TOST-based trends (in ppb/decade) of the mean ozone mixing ratios over the columns from the surface to 300 hPa,
TrOC (a), from 700 to 300 hPa, FTOC (b) and from surface to 700 hPa, LTOC (c) in the Northern Hemisphere during 1995-2021.**
**(d-f) The same as (a-c), but for relative trends (in %/decade). The statistics are based on QR median value of ozone mixing ratios in
each grid cell over 1990-2021. Significant trends are shown by a black dot in the grid cell.**



### 5.1.2 TrOC trends in 12 regions and 2 periods (1995-2021 and 2000-2021)

To better understand the difference between TrOC and FTOC trends in different regions and periods, we show regional trends
of TrOC and FTOC for two time periods (1995-2021 and 2000-2021), in both relative (Fig. 6 and Fig. 7) and absolute (Figs. S1 and S2) values. The trends with ± 2-sigma uncertainties and p-values for all the regions and periods are summarized in Table 4. The trends in 1995-2021 and 2000-2021 can indicate the variations of tropospheric ozone over the past three decades. We present the trends over 2000-2021 only for Northern South America and Malaysia/Indonesia, due to the shorter time series of available data there.

During 1995-2021, both TrOC and FTOC in East Asia and Southeast Asia show clear increasing trends, while the rest of regions have weak decreasing trends (Table 4). The TrOC trends are higher than the FTOC trends in Southeast Asia (diff = +0.8 ppb/decade or +0.2%/decade) and East Asia (diff = 0.3 ppb/decade or +1%/decade), suggesting that the increase in LTOC largely contributes to the increase of TrOC in these regions (Figs. 5 and S3). Other regions where the difference between TrOC and FTOC are larger than ±0.2 ppbv/decade include California, Southern USA, and Canadian Arctic. For the rest of the regions,
the difference between TrOC and FTOC trends is smaller than 0.2 ppbv/decade. During 2000-2021 (Table 4), TrOC in East Asia and Southeast Asia also increases, while the trends become slower in East Asia but faster in Southeast Asia. In the same period, the TrOC in the Malaysia and Indonesia region increases, with low uncertainty. The rest of the regions have weak decreasing trends with low certainty. The FTOC trends are generally similar to those of TrOC (Fig. 7c and 7d).

Comparing all the decreasing TrOC trends between the two periods (1995-2021 vs. 2000-2021), it appears that the decreasing
trends are stronger over 2000-2021 in more regions, including Northern Atlantic Ocean, European Arctic, Canadian Arctic, California, and East North America, but weaker over 2000-2021 in Southern USA, Western North America, and Continental Europe (Table 4). The FTOC trends between the two periods show similar variations in these regions. Among all regions with FTOC decreasing trends, the difference between the two periods appears largest in Canadian Arctic and European Arctic. Among the regions with increasing FTOC trends, East Asia is with a larger increase over 1995-2021 than over 2000-2021,
while Southeast Asia shows the opposite.

Figure 8a shows the TrOC and FTOC DLM estimates for the European TOST region in %/year since 1990. The difference between the two partial columns trends is clearly non-significant. However, both partial columns show a significant positive trend before 2010, and the trend estimates are significantly negative after 2015 for the FTOC. The consideration of a non-linear trend estimate allows highlighting periods of significance of the trend estimates. This is also made possible by the lower
uncertainty of the TOST product compared to the individual ozonesonde datasets. The same pattern can be observed for the North Atlantic Ocean region (Fig. 8b), and for Northeast Asia (Fig. 8c). In the latter case, the transition from positive to negative trend estimates occurs later in time due to a later reduction of NOx emissions in this region compared to other mid-latitudes (e.g. Gaudel et al., 2018). Note that the DLM behaviour at the beginning and end of the time series should not be over-interpreted, as those couple of years might be impacted by an artefact (dynamical model being less constrained, see Ball
et al., 2019).





**Figure 6: (a, b) TOST QR median trends (in %/decade) of the mean ozone mixing ratios from surface-300 hPa varying with latitude (left, a) and longitude (right, b), for the period of 1995-2021. Error bars represent the 95% confidence interval of the QR trend. Colors denote the regions. (c, d) The same as for (a, b), but for the period of 2000-2021. (e, f) The difference in the QR median trend between the periods of 1995-2021 and 2000-2021. Error bars represent the 95% confidence interval for the QR median trend difference.**







**Figure 7: The same as Fig. 6, but for the mean ozone mixing ratio over 700-300 hPa, FTOC.**




**Table 4. TOST-based TrOC and FTO3 trends ± 2-sigma uncertainties (2σ) and p-values (p-val) for 12 defined regions during 1995-2021 and 2000-2021. The bold and italic values indicate the trend is of high certainty (0.05 ≥ p > 0.01).**

| Region | 1995-2021, TrOC | | 1995-2021, FTOC | | 2000-2021, TrOC | | 2000-2021, FTOC | |
|---|---|---|---|---|---|---|---|---|
| | Trend ± 2σ | p-val | Trend ± 2σ | p-val | Trend ± 2σ | p-val | Trend ± 2σ | p-val |
| Eastern North America | -0.65±0.92 | 0.17 | -0.51±0.90 | 0.27 | -0.89±1.23 | 0.16 | -0.64±1.24 | 0.32 |
| Northern Atlantic Ocean | -0.26±0.64 | 0.43 | -0.16±0.84 | 0.70 | -0.57±0.95 | 0.25 | -0.65±1.32 | 0.34 |
| Canadian Arctic | -0.29±0.45 | 0.22 | 0.00±0.72 | 0.99 | -0.55±0.67 | 0.11 | -0.61±1.00 | 0.24 |
| Southern USA | -0.30±1.19 | 0.61 | -0.02±1.34 | 0.97 | -0.12±1.49 | 0.87 | 0.50±1.86 | 0.60 |
| Continental Europe | -0.32±0.71 | 0.37 | -0.44±0.87 | 0.32 | -0.24±0.88 | 0.60 | 0.03±1.10 | 0.95 |
| East Asia | *2.63±1.09* | *<0.01* | *2.31±1.24* | *<0.01* | *2.31±1.80* | *0.02* | 1.61±2.14 | 0.15 |
| Northern South America | / | / | / | / | -1.65±2.25 | 0.16 | -1.10±2.25 | 0.34 |
| Western North America | -0.26±0.57 | 0.36 | -0.31±0.68 | 0.37 | -0.13±0.88 | 0.78 | -0.10±1.00 | 0.85 |
| Malaysia/Indonesia | / | / | / | / | 0.66±3.26 | 0.69 | 1.43±3.18 | 0.38 |
| European Arctic | -0.29±0.45 | 0.22 | -0.13±0.73 | 0.73 | -0.41±0.67 | 0.23 | -0.68±0.81 | 0.11 |
| Southeast Asia | *2.67±1.01* | *<0.01* | *1.86±1.40* | *0.01* | *2.81±1.63* | *<0.01* | *2.74±1.83* | *0.01* |
| California | -0.31±1.16 | 0.59 | -0.61±1.03 | 0.24 | -0.55±1.42 | 0.45 | -0.74±1.36 | 0.29 |

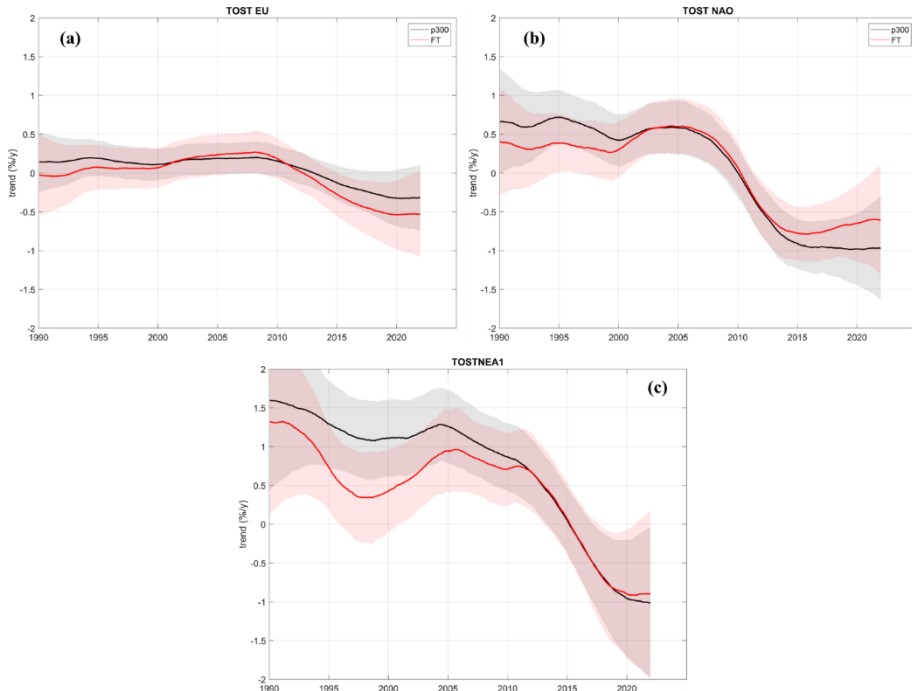

**Figure 8: DLM trends (%/yr) for both the TrOC (black) and FTOC (red) for the TOST regional time series in Europe (a), North Atlantic Ocean (b), and Northeast Asia (c). Shaded areas represent the 2σ uncertainties.**





### 5.1.3 Impact of the COVID-restrictions on the long-term trends

Satellite-based (Ziemke et al., 2022), ground-based and aircraft observations (ozonesondes, FTIR, IAGOS, e.g. Steinbrecht et al. 2021; Clark et al. 2021; Chang et al. 2022, 2023) reported negative ozone anomalies (−5%) during the COVID-19 in the free troposphere above western North America and Europe during 2020. Similar negative ozone anomalies were also detected at high elevation surface monitoring sites in Europe and North America (Putero et al., 2023). Model simulations of the COVID-19 period indicate that reduced emissions of ozone precursors across the Northern Hemisphere led to the ozone decreases (Miyazaki et al. 2020; Steinbrecht et al. 2021), reaching levels similar to those measured in the mid-1990s when ozone precursor emissions were less than 2019 levels (Chang et al. 2022). These low-ozone anomalies continued in the years thereafter, especially in (boreal) spring-summer at the northern mid-latitudes (Ziemke et al., 2022, Blunden and Boyer, 2024). Here, we compare the TrOC and FTOC trends up to 2019 (pre-COVID trends) with those up to 2021 (post-COVID trends) for the two periods (1995-2021 vs. 1995-2019 and 2000-2021 vs. 2000-2019) (Figs. 9 and S4 to S6). Figs. 9 and S4 show lower post-COVID trends than pre-COVID TrOC trends in all the regions. Especially, the TrOC trends changed from positive in pre-COVID to negative in post-COVID in Western North America and Continental Europe for both the periods. For FTOC, see Figs. S5 and S6, the post-COVID trends are also lower than those pre COVID in all the regions, and the reductions are larger for the period of 2000-2021 trend comparison. This confirms that the COVID restrictions have slowed down regional FTOC trends, especially in the more recent decades. Note that although the magnitude of trend differences between pre- and post COVID being small (<1 ppb/decade; <2%/decade), the signs of FTOC trends changed from positive in pre-COVID to negative in post-COVID in Western North America, California, and Northern Atlantic Ocean.

### 5.2 LMM synthesized trends

In this section, we will present the synthesized trends, calculated by the linear mixed-effects modelling approach, for the different regions whose boundaries are fixed by the CAMS TrOC correlation maps at the HEGIFTOM site locations. Trends are calculated for three different time periods (1990-2022, 1995-2022, and 2000-2022), for both the TrOC and FTOC amounts, and the impact of the COVID-restrictions on the trends will be assessed.

### 5.2.1 TrOC trends for different periods

In the upper panels of Fig. 10, we show the 1995-2022 TrOC trends and 2-sigma uncertainties for each region. If the 95% uncertainty ranges are in the same colour as the symbol of the trend value itself, the p-value of the trend is less than 0.05. From this plot and the values in Table 5, it can be seen that only the two Arctic regions have negative trends with very high confidence, around -1 ppb/dec (-1.5 %/dec). The Asian regions are associated with the strongest positive trends (between +1 and 6 ppb/dec, or +3 to more than 30 %/dec), most of them with medium confidence. The strongest TrOC increases took place in India and Southeast Asia (both around +6 ppb/dec or + 30 %/dec and +45 %/dec, respectively). For India, it should be noted that the time series extends only until 2018, with only 39% of the months having data (see Table 2). On the other hand, those





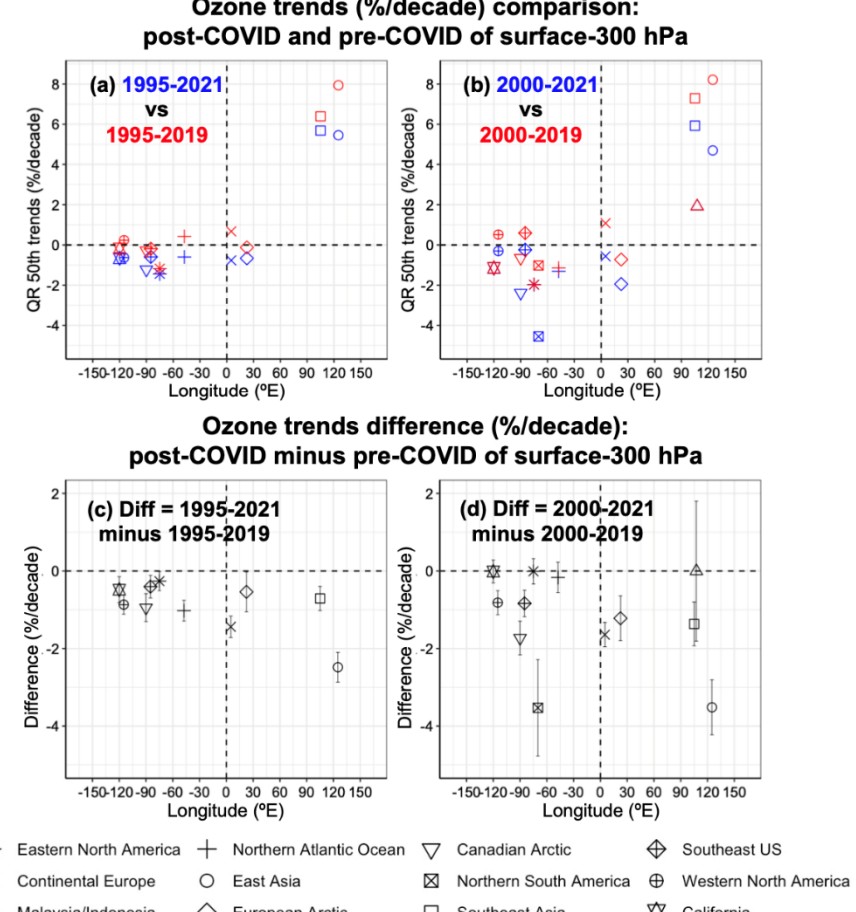

**Figure 9: (a) TOST QR median trends (in %/decade) of the mean ozone mixing ratio from surface-300 hPa varying with longitude, for the periods 1995-2021 and 1995-2019. The shapes denote the different regions. (b) The same as (a), but for the periods 2000-2021 and 2000-2019. (c) The difference in QR median trends between the periods 1995-2021 and 1995-2019. Error bars represent the 95% confidence interval for the QR median trend difference. (d) The same as (c), but now between the periods of 2000-2021 and 2000-2019. A version of this figure for the absolute trends (in ppb/decade) is available in Fig. S4.**

months have an average sampling rate of six measurements per month. For Southeast Asia, these numbers are respectively 29% and 11 measurements per month. Positive trends with high to very high confidence are also found for 3 North American regions: California (+ 2.2 ppb/dec, +5 %/dec), Western USA, and Southeast USA (both around +1 ppb/dec and 2 %/dec). It should be added here that Western USA incorporates the Californian sites, but also includes the three instruments in Boulder, Colorado. Inclusion of the Boulder data makes the regional trend less positive, but with higher confidence, compared to the Californian estimate. The other two American regions, Pacific Northwest and Eastern North America, have weak negative trend, with low confidence. A first preliminary conclusion might be that there is no uniformity of the trends over the North American continent. Other regions show a weak positive trend in Europe (medium confidence) and a positive trend in the Gulf of Guinea, also with medium confidence. Finally, when combining the trend estimates from different co-located techniques at







**Figure 10: Upper panels:** LMM TrOC synthesized 1995-2022 trends versus latitude (left) and longitude (right), calculated for the different regions, including only sites that have at least 30 monthly values. The regions are denoted by different symbols and colours for the trend values, and the error bars denote the 2-sigma uncertainties. If those 95% uncertainty ranges are in the same colour as the symbol of the trend value itself, the p-value of the trend is lower than 0.05; if shown in black, the p-value is higher than 0.05. **Middle panels:** same as in upper panels, but now for the 2000-2022 period. **Lower panels:** the LMM TrOC synthesized trends of the upper panels are shown now in black, and are compared with the LMM TrOC trends of the middle panels (in red).





Hawaii and Lauder (but extended with the FTIR at Wollongong, Australia), the resulting trends are positive (both around +0.5 ppb/dec and +1.5 %/dec), with very high confidence.

Let us now look at how consistent these patterns are over different time periods and we therefore consider the two most recent decades (2000-2022) (shown in middle panels in Fig. 10, values in Table 5 as well). It should be noted that the trends deviate no more than 1 ppb/dec (1.5 %/dec) between those two periods. We also observe that the 1995-2022 trends are greater than

the 2000-2022 trends in most regions, but only for the European Arctic significantly higher (see lower panels of Fig. 10). All North American and European regions and the African region have higher trends for the 1995-2022 period, whereas the trend increases or decreases for the more recent period are equally distributed over the 7 Asian regions. This finding is largely consistent with the continued decrease of emissions of ozone precursor substances (volatile organic compounds, nitrous oxides)

**Table 5: Synthesized TrOC trend estimates, 2-sigma uncertainties, p-values, and trend confidence for the different regions for 1995-**
**2022 (colums (1) to (3)), and for 2000-2022 (columns (4) to (6)). Following the TOAR-II statistical guidelines (Chang et al., 2023) the following degrees of certainty is assigned to a trend according to p value: very high certainty (p ≤ 0.01), high certainty (0.05 ≥ p > 0.01), medium certainty (0.10 ≥ p > 0.05), low certainty (0.33 ≥ p > 0.10), and very low certainty or no evidence (p > 0.33). Combining these uncertainty levels with the data coverage assignments in Table 2 gives the calibrated language for discussing confidence in long-term trend estimates as in Gaudel et al. (2024), and included here in columns (3) and (6).**

|  | (1) | (2) | (3) | (4) | (5) | (6) |
|---|---|---|---|---|---|---|
|  | 1995-2022 | | | 2000-2022 | | |
|  | Trend ± 2σ [ppb/dec] | p-value | confidence | Trend ± 2σ [ppb/dec] | p-value | confidence |
| Europe | 0.11±0.19 | 0.27 | medium | 0.11±0.24 | 0.34 | medium |
| Eur Arctic | -1.05±0.28 | <0.01 | very high | -1.80±0.37 | <0.01 | very high |
| Can Arctic | -1.00±0.46 | <0.01 | very high | -1.09±0.57 | <0.01 | very high |
| California | 2.20±0.78 | <0.01 | high | 1.75±0.77 | <0.01 | high |
| W USA | 0.84±0.55 | <0.01 | very high | 0.51±0.63 | 0.10 | high |
| Pacific NW | -0.06±0.53 | 0.83 | low | -0.22±0.66 | 0.51 | low |
| E N America | -0.19±0.41 | 0.36 | low | -0.85±0.49 | <0.01 | high |
| SE US | 1.07±0.70 | <0.01 | high | 0.79±0.84 | 0.06 | medium |
| E China | 4.36±2.06 | <0.01 | high | 3.00±2.74 | 0.03 | high |
| S Japan | 1.62±1.05 | <0.01 | medium | 1.69±1.33 | 0.01 | medium |
| NE Asia | 1.04±1.06 | 0.05 | high | 0.20±1.03 | 0.69 | low |
| SE Asia | 5.77±1.23 | <0.01 | medium | 2.86±2.25 | 0.01 | medium |
| S China Sea | 3.86±1.26 | <0.01 | medium | 2.72±1.61 | <0.01 | medium |
| S Malay Peninsula | 2.47±0.79 | <0.01 | medium | 1.97±0.87 | <0.01 | medium |
| India | 6.03±1.50 | <0.01 | medium | 6.15±2.25 | <0.01 | medium |
| Persian Gulf | 1.75±0.79 | <0.01 | medium | 1.60±0.79 | <0.01 | medium |
| Gulf of Guinea | 0.99±0.60 | <0.01 | medium | 0.88±0.67 | 0.01 | medium |
| Hawaii | 0.65±0.33 | <0.01 | very high | 0.81±0.44 | <0.01 | very high |
| Oceania | 0.53±0.27 | <0.01 | very high | 0.69±0.30 | <0.01 | very high |






in Northern America and Europe since the late 1980s, as opposed to East Asia, where the rapid reduction of NOx emissions in (some regions of) China started only since about 2011-2013, as observed by satellites (Gaudel et al., 2018, and references therein, Dufour et al., 2021). Meanwhile emissions have continued to increase in Southeast Asia (Li et al., 2024).

Starting 5 years earlier, the 1990-2022 TrOC trend values are only meaningful for a limited number of regions that therefore
rely on early ozonesonde and Dobson Umkehr time series (Europe, the Arctic regions, West USA, Lauder, and Hawaii). Other regions are largely dominated by the IAGOS measurements, that began in 1994. Of those 6 regions, only the Arctic regions show notable (and even strong for the Canadian Arctic) trend increases for the earliest period. Therefore, in the remainder of this paper, we will use 1995-2022 as our reference time period. Unless otherwise stated, those findings pertain to the 2 other considered time periods as well.

We conclude this section with the finding that the positive trends above Continental Europe and Northern America are diminished when shifting the beginning of the time period forwards, and that Arctic trends are progressively becoming more negative. The Asian trends do not show a consistent increase or decrease over the years. The positive trends at Hawaii and Oceania hardly change over the different time ranges considered.

### 5.2.2 Comparison of partial tropospheric ozone column trends

All TrOC regional trend findings described so far also apply for the regional free-tropospheric ozone column (FTOC, 700-300 hPa) trends. However, as the FTIR and Umkehr instruments are only sensitive to the entire tropospheric ozone column, FTOC retrievals are not available, and the Oceania and Hawaii "regions" have to be dropped from the FTOC analysis, because they are based on one (ozonesonde) time series only. To compare the contribution of the FTOC trends to the TrOC trends, we will compare their relative trends, as in Sections 5.1.1 and 5.1.2, the column-averaged FTOC mixing ratio being higher than its
TrOC counterpart (see Sect. 4).

The relative FTOC for the period 1995-2022 are shown in Fig. 11 and provided in Table S3. A direct comparison between the upper panel plots of Figs. 10 and 11 reveals that the general patterns are very similar, although both the metric (TrOC vs. FTOC) and the unit (ppb/dec vs. %/dec) differ. In the lower panels of Fig. 11, we show both the relative TrOC and FTOC trends. A first important point to note is that for almost all regions (except for Europe, India, and the Gulf of Guinea), the
relative TrOC and FTOC trends do not differ largely from each other (lying in each other's 2σ confidence intervals). If the relative trends of both the TrOC and FTOC have similar values, this means that the FTOC varies similarly as the TrOC, and the time variation of the boundary layer or lower-tropospheric ozone amounts are not expected to contribute majorly to the TrOC time variability or trend. And/or the impact of the FTIR and Umkehr time series, if available, on the TrOC trend, is rather limited. On the other hand, both effects could counterbalance each other. For the two Arctic regions, the TrOC and
FTOC trends are very similar, although they both include FTIR measurements from at least two instruments for their TrOC trend calculations. As the Arctic surface ozone trend is dependent from region to region and season to season (Law et al., 2023, Nilssen et al., 2024), it is hard to make strong conclusions here, although it seems that the lower troposphere does not affect the entire TrOC trends significantly. This is confirmed by the TOST analysis. For the Pacific Northwest, a region made up of



only IAGOS and ozonesonde measurements, the nearly identical TrOC and FTOC trends really point to a limited impact of

the lower-tropospheric ozone trend on the entire tropospheric column.

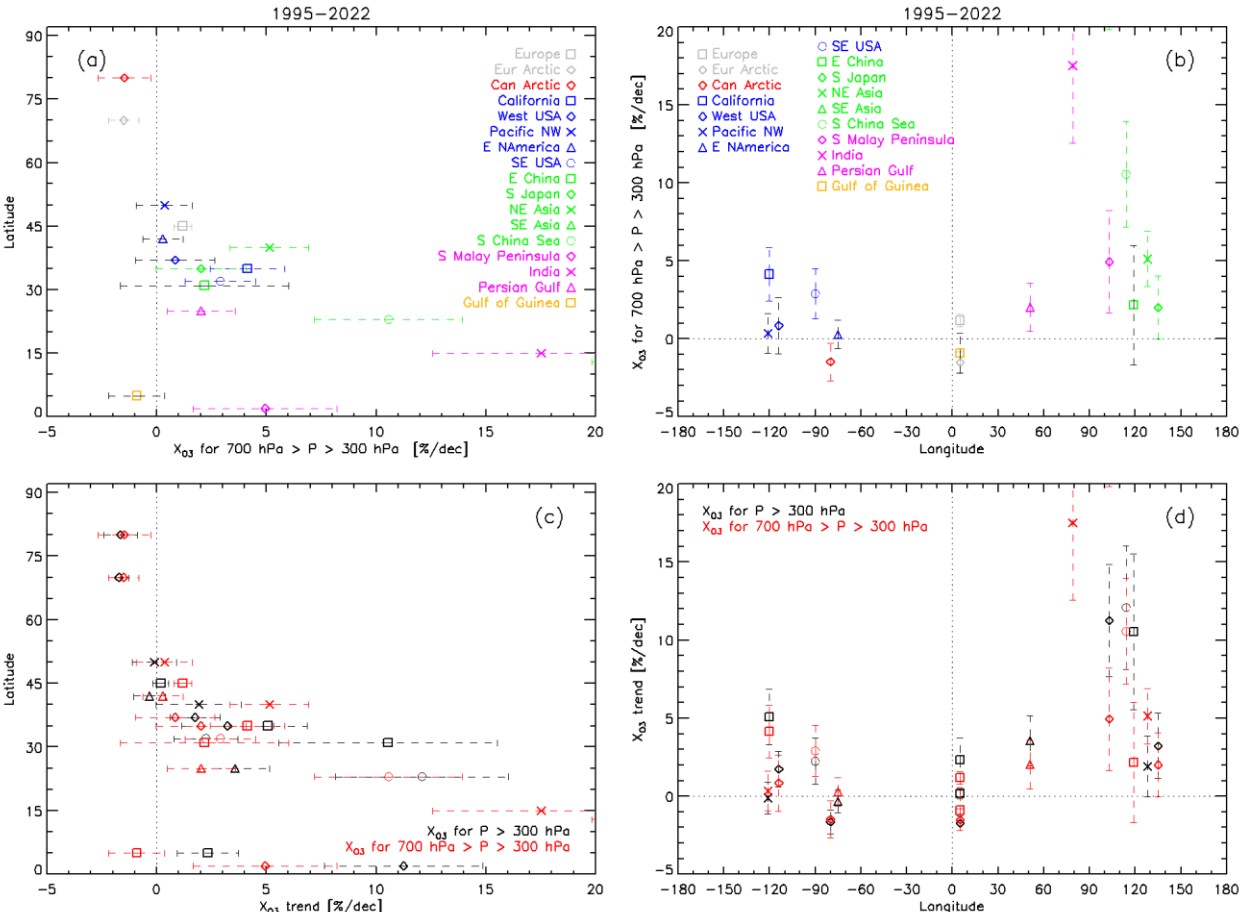

**Figure 11: Upper panels: Same as in upper panels of Fig. 10, but now for the relative FTOC trends (%/dec) for the 1995-2022 period. Lower panels: the LMM relative FTOC synthesized trends of the upper panel are shown now in red, and are compared with the**
**LMM relative TrOC trends for the same period 1995-2022 (in black). Note that the relative TrOC trends for SE Asia (~45 %/dec) and India (~30 %/dec), and the relative FTOC trend for SE Asia (~ 25%/dec) are missing in the plots, to increase the visibility for the trend estimates and their differences for the other regions.**

Overall, there is an almost equal distribution of regions with higher relative TrOC than FTOC trends than the other way around.

Regions with higher relative TrOC than FTOC trends are California and Western USA regions (which are basically the same

for the FTOC, except for the inclusion of the ozonesonde time series in Boulder for the latter), the Gulf of Guinea, and all

Asian regions except Northeast Asia. This finding is not surprising, as the Gulf of Guinea and Asian regions have both the

largest ozone mixing ratios and the strongest ozone increases in the lower troposphere (see Wang et al., 2022; Gaudel et al.,

2020, 2024, Stauffer et al., 2024), so that the stronger TrOC trends are mainly driven by the LTOC trends. The impact of the

short FTIR time series in the TrOC trends of East China (Hefei) and South Japan (Tsukuba) should be rather limited. For





Northeast Asia, the long and dense FTIR Rikubetsu and shorter but dense FTIR Moshiri time series affect the TrOC time series, making its synthesized trend higher than the FTOC trend, based on IAGOS sites only. For the California and Western USA regions the origin of the higher relative TrOC than FTOC is less understood, in particular taking into account that Chang et al. (2023) identified boundary layer decreases versus free-tropospheric increases for a similar Western North America fused dataset. A significant negative lower-tropospheric (surface to 700 hPa) ozone trend is also observed from the TOST data in

this region, see Sect. 5.1.2. Transport of tropospheric ozone from Asia impacts the ozone trends in Western United States, e.g. Verstraeten et al. (2015) estimated this transport to have offset 43% of the expected reduction in FTOC there.

Finally, in a couple of regions (Europe, Pacific Northwest, Southeast US, and Eastern North America), the relative FTOC trends are higher than the TrOC trends. In those regions, the impact of the FTIR and Umkehr time series is non-existent (Pacific Northwest, Southeast US) or thought to be minimal. Depending on the study, the vertical profile of tropospheric ozone trends

in Europe based on (merged/fused) IAGOS and ozonesonde measurements is rather flat (Figs. 5 and 6 in Wang et al., 2022) or shows a maximum in the boundary layer (Fig. 5 in Chang et al., 2022). Gaudel et al. 2020 reported that median LTOC above Europe increased at a rate similar to the FTOC rate, while median LTOC above the Southeast US and Eastern North America regions is largely unchanged over the 1994–2016 period. The authors conclude that, while precursor reductions have been effective at reducing the high ozone events at the surface and in the lower troposphere, mainly in summer, the annual TrOC

trends above North America and Europe are largely positive, driven by the free tropospheric ozone increases. Our study seems to align with those conclusions.

As already mentioned, the hereabove described regional patterns in relative TrOC vs. FTOC trends are also valid for the 1990-2022 and 2000-2022 time periods too.

### 5.2.3 Impact of the COVID-restrictions on the long-term trends

In Van Malderen et al. 2024, we already assessed the impact of the post-COVID period on the trends at individual sites; here we will compare the regional LMM synthesized pre-COVID trends (up to 2019) with the post-COVID trends (up to 2022), as in Sect. 5.1.3. In Fig. 12, the synthesized 1995-2019 TrOC trends are shown for all regions and compared with the 1995-2022 trends. The pre-COVID trend values and uncertainties are provided in Table S2 in the supplement. It is striking that for all regions (except Hawaii, SE Asia: no change) the 1995-2019 trends are greater than the 1995-2022 trends, and much higher for

the European Arctic and (Continental) Europe regions (p< 0.05; high confidence). The Indian region is omitted from this discussion as the data only extend through 2018. Trend reductions are typically -1 ppb/dec (-2%/dec), except for South Japan and Southern Malay Peninsula (~ -5 to -7 %/dec, but with large trend uncertainties). We found similar results for the trends calculated for the periods starting in 1990 and 2000. Similarly, the FTOC trends ending in 2019 are greater than for the trends ending in 2022 for almost all regions (see Table S3 for trends calculated from 1995), with again significance for especially

Europe and the European Arctic. But the FTOC trend differences between both periods (in Table S3) are smaller, with basically the same FTOC trend values for East China, South Japan, NE Asia, SE Asia, and Pacific NW, West and Southeast USA. The FTOC trend reductions lie again between 0 and -1 ppb/dec (-2%/dec), except for South China Sea and Southern Malay





Peninsula (-4%/dec). Our analysis hence confirms that the COVID restrictions have slowed down the decades long positive regional TrOC and FTOC trends, with the most significant trend reduction over Europe, and to a lesser extent Eastern North America. At the same time, the decreasing Arctic TrOC and FTOC trends (with the exception of the positive Canadian Arctic FTOC 1990-2022 trends) have (significantly for European Arctic) turned even more negative with the inclusion of the post-COVID-19 period.

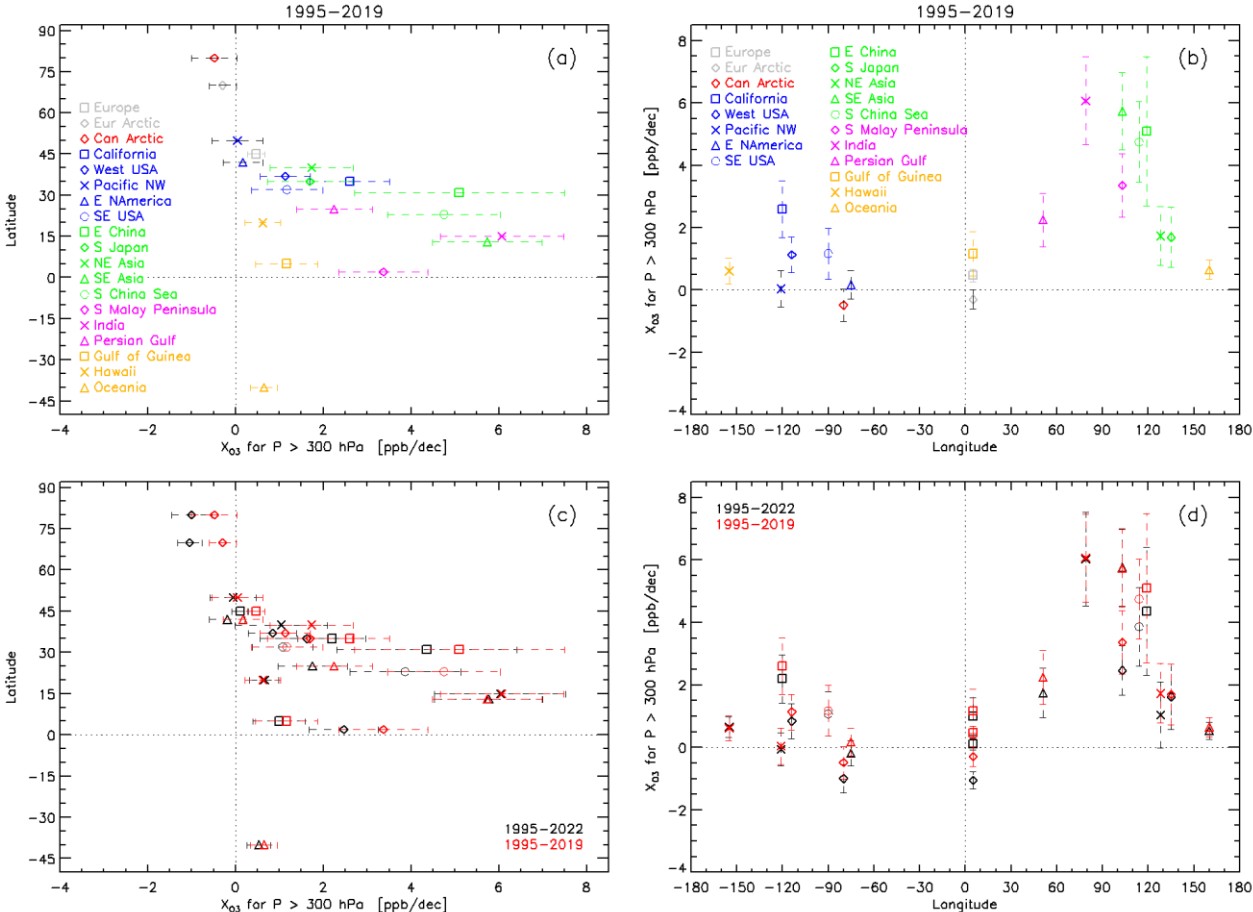

**Figure 12: Upper panels: Same as in upper panels of Fig. 10, but now for the TrOC trends for the 1995-2019 time period, hence excluding the period where restrictions have been applied to cope with the COVID-19 spread. Lower panels: the synthesized trends of the upper panel are shown now in red, and are compared with the TrOC trends for the time period 1995-2022 (in black).**

## 5.3 Comparison of regional trends

In this section, we compare the regional trends calculated from TOST and the LMM regional synthesized trends from the entire HEGIFTOM dataset. We should make two important considerations first. Although the determination of the regions was mainly driven by the correlation analysis described in Section 3.1, the regions used for both approaches differ. The main reason is the optimization of the regions to have enough temporal sampling within a region. Therefore, the TOST regions, based only



on ozonesonde measurements, are in general larger than the LMM regions. There are also a number of non-overlapping regions
(e.g. Hawaii and Australia for LMM, Northern Atlantic Ocean and Northern South America for TOST). A second point we
want to address is that TOST is based on a larger sample of ozonesondes sites than the homogenized HEGIFTOM ozonesonde

dataset used for LMM trend calculation. Notable examples are the non-homogenized Japanese ozonesonde sites which form
the backbone for the East Asia TOST trends, but have not been included in HEGIFTOM database and, as a consequence, the
LMM trend calculation. Christiansen et al. (2022) reported a potential step change occurring at these sites around 2010 in the
troposphere, when the ozonesonde type changed.

In Fig. 13, the regional TrOC trends since 2000 are shown in colours, with open symbols denoting the LMM trends, and closed

symbols the TOST trends. We first consider the 2000-2021/22 period, as only this period has been used for calculating TrOC
trends from the individual sites in Van Malderen et al. (2024), which are shown in the background of Figure 13 as well.

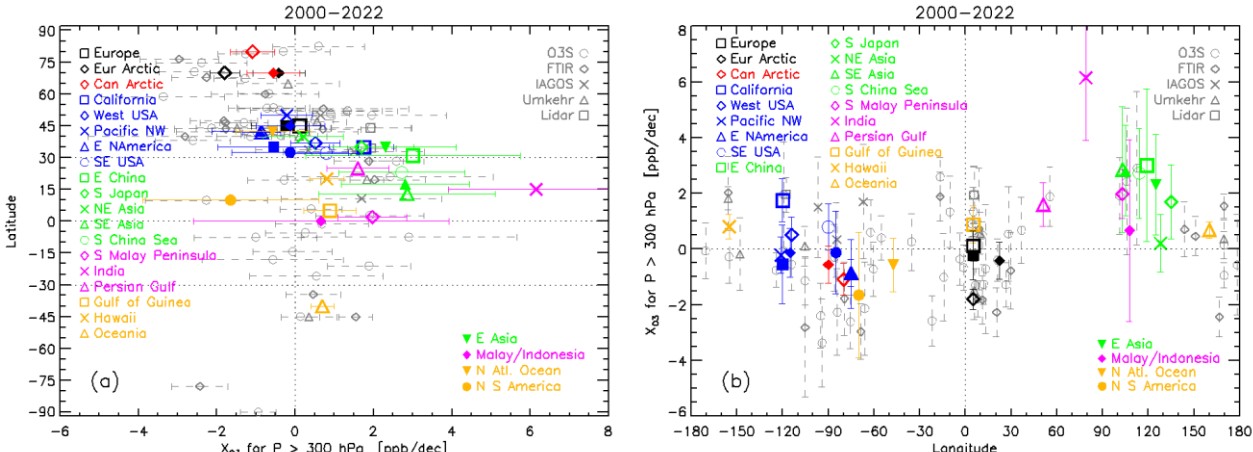

**Figure 13: All 2000-2022 TrOC trends [ppb/dec] estimated from the homogenized ground-based and in-situ HEGIFTOM measure-**
**ments. In grey: the individual site trend estimates, with different symbols for the different techniques. Open colored symbols denote**
**the LMM trend estimates for the different regions, marked with the different colours in the upper left legends in the plot. The closed**
**symbols represent the TOST regional trend estimates, using the same colour coding for the common regions as for the LMM. Non-**
**overlapping TOST regions are identified in the lower left legends in the plots.**

Both TOST and LMM reveal positive trends in the Asian regions, so this is a strong feature present in the time series of all the

different techniques, not only ozonesondes. Also the negative Arctic trends are a common feature of both regional trend
estimation techniques. This was also the only consistent geographical pattern that could be derived from the individual site
trend estimates. The TOST trends are however less negative than the LMM trends, the latter being impacted by FTIR time
series with strong negative trends (see Van Malderen et al., 2024). In the other common regions, mainly Europe and North
America, the TOST trend estimates are always smaller (and negative) than the LMM trend estimates, although the differences

are small (and the trend estimates lie even on top of each other for Eastern North America in Fig. 13). An exception is the
California regions, in which trends are strongly positive for LMM and negative for TOST. However, the TOST and LMM
sampling strategies are very different for this region, with the TOST trends based almost entirely on the single ozonesonde





record at Trinidad Head, with negative TrOC trends. The LMM dataset is based on several monitoring sites, but not Trinidad Head (low TrOC correlation with the other sites), and the strong LMM trend is dominated by the dense and accurate (low uncertainty) TMF Lidar time series. In general, it seems that in Europe and North America, the inclusion of other techniques is responsible for higher LMM trends compared to TOST. When comparing the individual site trends with the regional trends at a glance, we can conclude that the regional trends from both approaches seem to capture quite well the "mean" variability displayed by the individual site trends.

Finally, our findings with respect to the comparison of the regional TOST and LMM trends persist when considering the 1995-2021/22 period, as shown in Fig. S7. The common findings for TOST and LMM in terms of variability of the trends when considering different beginning years (1990, 1995 or 2000), relative contribution of the FTOC to the TrOC trends, and the impact of the post-COVID period are summarized in the conclusions below.

## 6 Conclusions

In this paper, we exploit the homogenized ground-based and in-situ tropospheric ozone measurements from the TOAR-II Focus Working Group HEGIFTOM (Harmonization and Evaluation of Ground-based Instruments for Free-Tropospheric Ozone Measurements) to calculate regional trends of (partial) tropospheric ozone columns. In a separate HEGIFTOM paper, Van Malderen et al. (2024), those trends have been presented for the individual time series at 55 sites, revealing a more or less equal amount of positive and negative trends, at sites around the world that are not evenly distributed. Here, we wanted to explore if a more consistent understanding of the geographical distribution of ozone trends can be obtained by focusing on regional trends, which increase the monthly sampling frequency w.r.t. individual time series, and which might include shorter time series from sites or time series with gaps. This initiative required the development of three novel approaches. First, the regions for which trends are estimated are determined with a correlation analysis between the tropospheric ozone column monthly anomalies retrieved from CAMS. Secondly, whereas the Trajectory-mapped Ozonesonde dataset for the Stratosphere and Troposphere (TOST) has already been used to calculate mean latitudinal trends for tropospheric ozone columns (Gulev et al., 2021), it is the first time that it is used for calculating regional trends. Thirdly, a linear mixed-effects modelling approach is used to calculate synthesized trends in the defined regions from the homogenized HEGIFTOM individual site trends, for time series that have at least 30 monthly values.

The most important findings are:

- For the different considered periods (1990-2021/22, 1995-2021/22, 2000-2021/22), both approaches give increasing (partial) tropospheric ozone column amounts over almost all Asian regions, and negative TrOC and FTOC trends over the Arctic regions. Trends over Europe and North America are mostly weakly positive for LMM, but weakly negative for TOST. The difference is, at least for some regions, driven by the positive trends from measurement techniques other than ozonesondes. For the non-common regions, we note that TOST reveals negative trends over





the Northern Atlantic Ocean and Northern South America, while LMM displays positive trends over the Gulf of
       Guinea, the Persian Gulf, Hawaii and Australia.

- In terms of the relative contributions to the tropospheric ozone column trends, TOST and LMM agree on the following
  conclusions. The TrOC increases over the Asian regions are higher than the corresponding (relative) FTOC increases
  and are mainly due to the strong increases in lower-tropospheric ozone column trends. Also in the west of North
  America, the TrOC trends are greater than the FTOC trends. In the other regions, the TrOC trends are slightly lower

or similar than the FTOC trends, which is clearly due to decreasing lower tropospheric ozone column trends in e.g.
       the eastern regions of the USA.

- For both approaches, the 2000-2021/22 trends decreased in magnitude compared to the 1995-2021/22 for most of the
  regions.

- The pre-COVID trends are for all regions larger than the post-COVID trends. This is a very consistent feature for

both approaches, and for all periods.

It should be noted that only annual median trends have been treated here, although the trends may vary from season to season
or from percentile to percentile. The variation of seasonal or low/high percentile trends could be quite different from the annual
trends described here. Further investigations of tropospheric ozone trends might consider the impact of specific months or
seasons or percentiles, to aid in identifying the drivers of the long-term annual trends. A necessary condition for deriving such

trends is a high (enough) monthly sampling frequency, which should be explored first for the ground-based and in-situ
observations.

While we have provided the widest possible distribution of regional-scale ozone trends around the world, the global coverage
is still limited. Most regions are clustered in North America, Europe and Asia and only one region is located fully in the
southern hemisphere. Despite their spatial limitations, the geographical distribution of the (partial) tropospheric ozone trends

and the relative contributions of the free-tropospheric and low-tropospheric ozone column trends to the TrOC trend are a
valuable dataset for validating satellite TrOC trends with global coverage. Satellite tropospheric ozone column retrievals often
lack the vertical sensitivity information that the ground-based and in-situ datasets provide. Our results may also help to assess
the performance of atmospheric chemistry models to accurately represent the distribution and variation of tropospheric ozone
columns, provided that the model output is temporally and spatially matched to the ground-based and in-situ observations.

**Appendix: Impact of sites with limited data availability on the LMM trends**

The LMM approach for calculating synthesized trends does take into account the trends for individual sites in the regression
model, although there are site-specific adjustments to the synthesized trend. Sites with a small number of measurements, either
in terms of sampling frequency within a month or a rather limited time series due to gaps or the absence of a long-term
monitoring program, contribute only marginally to the synthesized trend. However, to assess the impact of short-term time

series on the LMM approach, synthesized trends were calculated including all sites, and including sites with at least 30, 60,





and 120 months of data available. The removal of sites with sparse monthly data, which are predominantly IAGOS airports, has consequences for the percentage of months with data, and the average number of TrOC measurements in those months. Those statistics are most heavily impacted in the regions that are already most sparsely sampled, making it impossible to calculate synthesized trends for East China, Northeast Asia, SE Asia, India, Indonesia, Persian Gulf, East Mediterranean Sea,

Gulf of Guinea, Middle America, if at least 120 monthly values per site are required. Therefore, this criterion seems to be too strict. If at least 60 monthly values per site are required, the percentage of months with data is halved in two regions (East China and SE Asia), thereby changing the beginning (2009, East China) and ending year (2006, SE Asia) of the time periods, disabling trend detection. Requiring at least 60 monthly values per site also decreased the average number of TrOC measurements in the months with data to less than 10 in the regions South China Sea, Persian Gulf, and Gulf of Guinea. But

trend detection is still possible here, with low confidence. On the other hand, as the synthesized trend detection with the 60 monthly values criterion is basically based on the time series of only one site in the regions Indonesia (O3S Watukosek), East Mediterranean Sea (IAGOS TLV), and Central America (O3S Costa Rica), these "regional" trend estimates should be discarded as well. The latter observation also holds when limiting the LMM trend estimation to sites that contain at least 30 months of available data. However, with this criterion, the East China and SE Asia, although containing large gaps (resp. 40

and 30% of the months covered with data), extend over the entire 1995-2021/22 period, with high temporal sampling in the months with data available (resp. ~25 and 11 observations a month, see Table 2, the former especially due to the dense FTIR Hefei time series).

If we now look at the synthetized TrOC trend differences for 1995-2022 for the LMM variants without any restriction on sites and with only including sites with at least 30 and 60 monthly values (see Fig. A1), it should be noted that trend differences are

marginal, in particular between the LMM variants without any restriction and with only sites with at least 30 monthly values (upper panels). The TrOC differences between the LMM variants with including only sites with at least 30 or 60 monthly values are larger (but still less than 1 ppb/dec, see lower panels of Fig. A1) for, not surprisingly, the more sparsely sampled Asian regions (Southern Malay Peninsula, South China Sea, and Northeast Asia). In those three regions, the sample reduction also leads to a trend reduction (of around 1 ppb/dec), and increases the trend confidence level for Northeast Asia.

More in general, we find that the synthesized trend differences for the different lower limits of available months are for all considered periods less than 1 ppb/dec (5%/dec) for both TrOC and FTOC. We also mention here that the trend uncertainties (both the standard deviations, see Fig. A1 for the TrOC 1995-2022 trend 2σ values, and the p-values) are very similar for the different LMM variants.

We can therefore conclude that applying the LMM trend estimation for sites that have at least 30 monthly values available

seems to be a good compromise between limiting the impact of trends at individual sites with a very limited amount of data and retaining a useful number of regions.


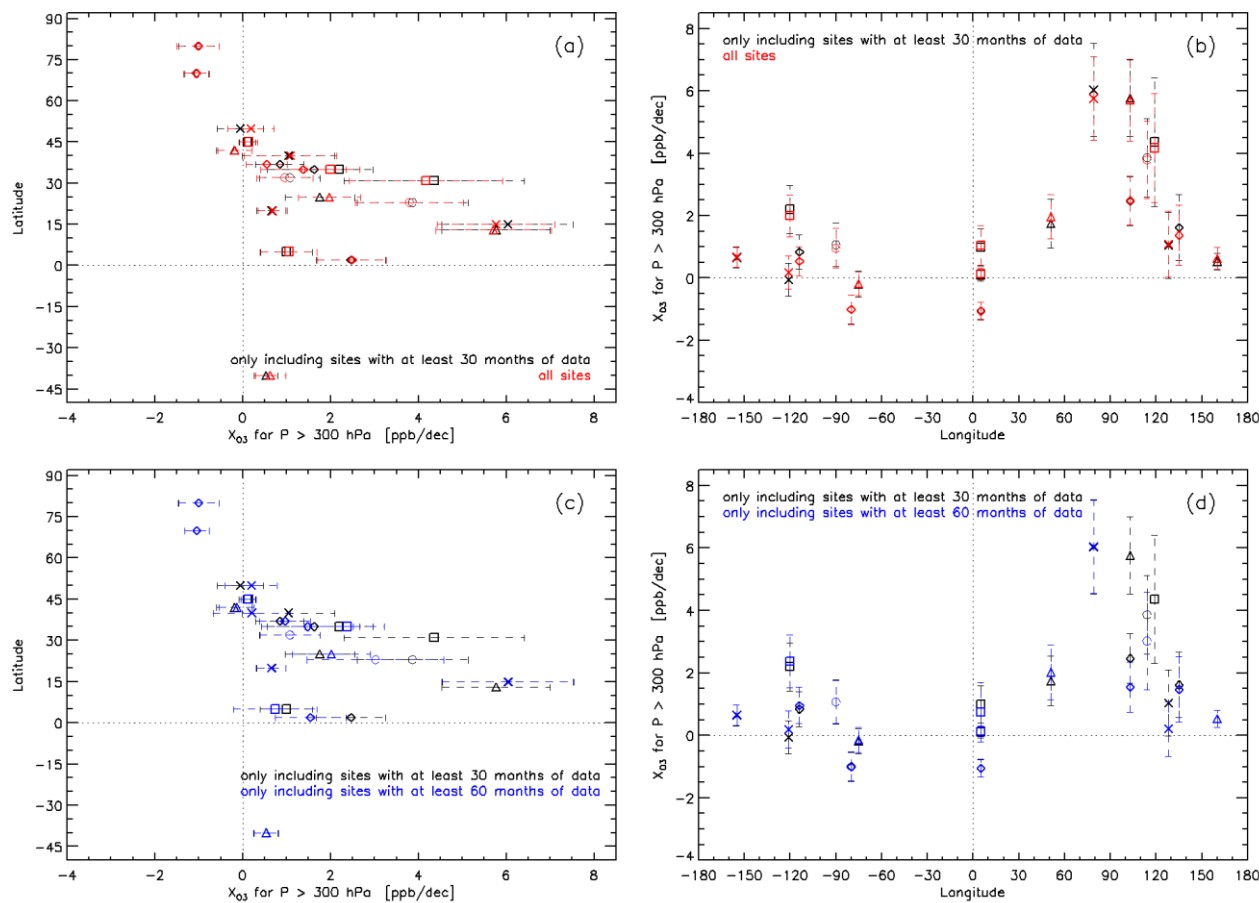

**Figure A1: Upper panels: LMM TrOC trend estimates for the 1995-2022 period, only including sites with at least 30 months of data (in black), and without excluding any sites (in red), versus latitude (a) and longitude (b). Lower panels: same as the upper panels, but now only including sites with at least 60 months of data (in blue). Note that there are in panels (c) and (d) two regions (East China and SE Asia) for which no trends can be calculated when including only sites with at least 60 months of data (see text).**

## Author contributions

Conceptualization & Methodology: RVM, ZZ, KLC, RB, ORC, JL, EMB, CV, IP, TL, VT, AG, DWT; Formal Analysis & Visualization: RVM, ZZ, KLC, RB, ORC, JL, EMB; Data Curation: RVM, EMB, CV, IP, TL, VT, PW, PE, DWT, HGJS, AMT, RMS, DEK, DP, GA, MRDB, MMF, JWH, JLH, BJJ, NJ, RK, IM, IM, GM, KM, IM, JN, AP, MP, RQ, VR, DS, WS, KS, RS; Writing – original draft preparation: RVM, ZZ, KLC, RB, ORC, JL, EMB; Writing – review & editing: all.



**Competing interests**

The authors declare that they have no conflict of interest.

**Acknowledgements**

**TO BE FURTHER COMPLETED!** The FTIR monitoring program at Jungfraujoch was primarily supported by the F.RS. - FNRS (Brussels, Belgium) and the GAW-CH program of MeteoSwiss (Zürich, Switzerland). Peter Effertz and Irina Petropavlovskikh's research was supported by an NOAA Cooperative Agreement with CIRES, NA17OAR4320101. The National Center for Atmospheric Research is sponsored by the National Science Foundation. The NCAR FTS observation programs at Thule, GR, Boulder, CO and Mauna Loa, HI are supported under contract by the National Aeronautics and Space
Administration (NASA). The Thule work is also supported by the NSF Office of Polar Programs (OPP). We wish to thank the Danish Meteorological Institute for support at the Thule site and NOAA for support of the MLO site. We are indebted to all the instrument operators, station (co-)PIs, and funding agencies without which the ozone data records used for this research would not have been available. The data used in this publication were obtained as part of the Network for the Detection of Atmospheric Composition Change (NDACC) and are available through the NDACC website https://www.ndacc.org. Original
ozonesonde data are also stored at https://www.woudc.org and https://tropo.gsfc.nasa.gov/shadoz/. MOZAIC/CARIBIC/IAGOS data were created with support from the European Commission, national agencies in Germany (BMBF), France (MESR), and the UK (NERC), and the IAGOS member institutions (http://www.iagos.org/partners). The participating airlines (Lufthansa, Air France, Austrian, China Airlines, Hawaiian Airlines, Air Canada, Iberia, Eurowings Discover, Cathay Pacific, Air Namibia, Sabena) supported IAGOS by carrying the measurement equipment free of charge
since 1994. The data are available at http://www.iagos.fr thanks to additional support from AERIS.

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
