# Peer review of "Ground-based Tropospheric Ozone Measurements: Regional tropospheric ozone column trends from the TOAR-II/ HEGIFTOM homogenized datasets"

_EGUsphere, 2024_

## Community Comment (CC1)

February 5, 2025

**Comments by Gabriele Pfister (TOAR Steering Committee Member) on:**

Van Malderen, R., Zang, Z., Chang, K.-L., Björklund, R., Cooper, O. R., Liu, J., Maillard Barras, E., Vigouroux, C., Petropavlovskikh, I., Leblanc, T., Thouret, V., Wolff, P., Effertz, P., Gaudel, A., Tarasick, D. W., Smit, H. G. J., Thompson, A. M., Stauffer, R. M., Kollonige, D. E., Poyraz, D., Ancellet, G., De Backer, M.-R., Frey, M. M., Hannigan, J. W., Hernandez, J. L., Johnson, B. J., Jones, N., Kivi, R., Mahieu, E., Morino, I., McConville, G., Müller, K., Murata, I., Notholt, J., Piters, A., Prignon, M., Querel, R., Rizi, V., Smale, D., Steinbrecht, W., Strong, K., and Sussmann, R.: *Ground-based Tropospheric Ozone Measurements: Regional tropospheric ozone column trends from the TOAR-II/ HEGIFTOM homogenized datasets*, EGUsphere [preprint], https://doi.org/10.5194/egusphere-2024-3745, 2025.

This review is by Gabriele Pfister, member of the TOAR-II Steering Committee. The primary purpose of these reviews is to identify any discrepancies across the TOAR-II submissions, and to allow the author teams time to address the discrepancies. Additional comments may be included with the reviews. While members of the TOAR Steering Committee may post open comments on papers submitted to the TOAR-II Community Special Issue, they are not involved with the decision to accept or reject a paper for publication, which is entirely handled by the journal's editorial team.

**Comments regarding TOAR-II guidelines:**
TOAR-II has produced two guidance documents to help authors develop their manuscripts so that results can be consistently compared across the wide range of studies that will be written for the TOARII Community Special Issue. Both guidance documents can be found on the TOAR-II webpage: https://igacproject.org/activities/TOAR/TOAR-II The TOAR-II Community Special Issue Guidelines: In the spirit of collaboration and to allow TOAR-II findings to be directly comparable across publications, the TOAR-II Steering Committee has issued this set of guidelines regarding style, units, plotting scales, regional and tropospheric column comparisons, and tropopause definitions.
The TOAR-II Recommendations for Statistical Analyses: The aim of this guidance note is to provide recommendations on best statistical practices and to ensure consistent communication of statistical analysis and associated uncertainty across TOAR publications. The scope includes approaches for reporting trends, a discussion of strengths and weaknesses of commonly used techniques, and calibrated language for the communication of uncertainty. Table 3 of the TOAR-II statistical guidelines provides calibrated language for describing trends and uncertainty, similar to the approach of IPCC, which allows trends to be discussed without having to use the problematic expression, "statistically significant".

General Comments

The study uses the homogenized ground-based and in-situ tropospheric ozone measurements from the TOAR-II Focus Working Group HEGIFTOM (Harmonization and Evaluation of Ground-based Instruments for Free-Tropospheric Ozone Measurements) to conduct an in-depth analysis of regional trends of total and partial tropospheric ozone columns over the 1995-2022 time period. It builds on the previous Van Malderen et al. (2024) analysis which presented trends for the individual sites based on the HEGIFTOM data set.

The overarching objective of TOAR-II is to provide an updated state of science of the ozone global distribution and trends relevant to climate, human health and vegetation. As such this paper makes an important contribution to the TOAR-II objective and also provides valuable input for evaluating chemical transport models as well as guidance on calculating trends.

The study is very detailed and carefully conducted and limitations in both the measurements and statistical analysis are characterized. The authors introduce new approaches to define regions for the trend calculation (a re-analysis based correlation analysis), include the Trajectory-mapped Ozonesonde dataset for the Stratosphere and Troposphere (TOST) in their analysis for comparison and present a linear mixed-effects modelling approach to calculate synthesized trends. Where possible, the authors compare their calculated trends to previous studies and if discrepancies arise, they are investigated and if possible explained.

While the trends in many parts could be explained by changing precursor emissions, I think the results do give motivation for further studies to understand what other drivers besides emissions might be contributing to the trends.

Specific Comments

The paper requires careful reading because of the amount of detail discussed and the different advanced statistical methods involved. It would help for understanding to make more clear what the different methods were used for and why the different methods, specifically as it relates to LMM and DLM. Otherwise readers who are not fully familiar with advanced statistics might find it challenging to follow the paper.

The figures contain a lot of information and this makes them hard to read. Maybe there is a way to simplify some of them and keep the current complex figures in a supplement. E.g.,:
Figure 6 & 7: It is not clear to me what the difference between filled and open symbols is. Maybe it would help to discern the different regions (which I found challenging because some of the colors look very similar) and the region-specific colors and symbols could be used for all the plots in a consistent manner (at the moment squares are used for median trends and circles for the difference plots).
Figure 8: Increase the font size for the labels
Figure 10&11&12: Maybe increasing the font size and making the symbols larger could help with the readability of the graphs?
Figure 3: It might be of interest to also include sites that were not used in the analysis in a different symbol.

---

## Author Comment (AC1)

**Response to Referee #1 comment on "Ground-based Tropospheric Ozone Measurements: Regional tropospheric ozone column trends from the TOAR-II/ HEGIFTOM homogenized datasets" by Van Malderen et al.**

The manuscript has presented the regional mean tropospheric ozone trends using the currently available homogenized ground-based and in-situ tropospheric ozone measurements, as a research outcome from the TOAR-II HEGIFTOM working group. Several statistical methods, including spatial correlation analysis, quantile regression, and linear mixed-effects modeling, are applied to test the consistency of tropospheric ozone trends as estimated from different regions, periods and datasets. Overall, the study is a valuable addition to our current understanding by summarizing the free tropospheric ozone trends revealed by HEGIFTOM datasets. Such datasets and estimated trends will provide benchmarks for model evaluation.

Thank you very much for your positive feedback and for taking your time to review the manuscript. Your suggestions have been taken into account carefully and we believed they improved the manuscript substantially.

Here I have several comments that the authors should address to further improve the manuscript.

**Specific Comments**

1) Page 2, Line 58-59, Abstract –

Here, median confidence and high confidence were used to describe the robustness of the trends; however, how were these terms defined in the main text? Some explanation was mentioned in Table 5, yet I think some more details are needed to clarify their statistical meaning.

You are right, the explanation of this concept deserves to be in the main text as well. As a matter of fact, the use of terms such as median confidence and high confidence can be traced back to IPCC AR5, as described by Mastrandrea et al. (2010, 2011). So, we are just following well-known IPCC methods and we refer the reader to these papers. In the beginning of Section 5.1.2, we added:

"Table 5 lists the trend estimates, 2-sigma uncertainties, $p$-values, and trend confidence. We hereby use the uncertainty scale for assessing the reliability and likelihood of the estimated trend that has been developed within TOAR-II (Chang et al., 2023b): very high certainty ($p \leq 0.01$), high certainty ($0.05 \geq p > 0.01$), medium certainty ($0.10 \geq p > 0.05$), low certainty ($0.33 \geq p > 0.10$), and very low certainty or no evidence ($p > 0.33$). Following the methodology of the IPCC Assessment Report 5 (Mastrandrea et al. (2010, 2011), we adopt the concept of trend confidence, which combines this trend uncertainty (based on the $p$-value and the 95% confidence interval) with the data coverage, as recently applied by Gaudel et al. (2024), based on the number of observations per month and continuity of sampling (see Table 2 for the classification). For example, higher confidence trends will be obtained for trends with lower $p$-values and higher data coverage. Table A1 in Gaudel et al. (2024) provides the translation table or calibrated language for assigning the confidence level based on trend uncertainty and data coverage, and has been applied here."

Mastrandrea, M. D., Field, C. B., Stocker, T. F., Edenhofer, O., Ebi, K. L., Frame, D. J., Held, H., Kriegler, E., Mach, K. J., Matschoss, P. R., and Plattner, G. K.: Guidance note for lead authors of the IPCC Fifth Assessment Report on consistent treatment of uncertainties, Intergovernmental Panel on Climate Change, https://www.ipcc.ch/site/assets/uploads/2017/08/AR5_Uncertainty_Guidance_Note.pdf (last access: 13 May 2025), 2010.

Mastrandrea, M.D., Mach, K.J., Plattner, G. K., Edenhofer, O., Stocker, T. F., Field, C. B., Ebi, K. L., and Matschoss, P. R.: The IPCC AR5 guidance note on consistent treatment of uncertainties: a common approach across the working groups. Climatic Change 108, 675, https://doi.org/10.1007/s10584-011-0178-6, 2011.

2) Page 7, Line 225 -

Twenty-four well-correlated regions were identified here using the spatial correlation analysis; however, it was unclear how these 24 regions were located and their covered areas. Such information was unclear by looking at Figure 1 as an example.

In Gaudel et al. (2020) and Wang et al. (2022), see their Fig. 1, 11 different regions are proposed in which IAGOS and ozonesonde tropospheric ozone profile data are merged based on co-location. For the airports and ground-based sites located within those regions, we performed the correlation analysis between the CAMS TrOC/FTOC time series at those locations. We then refined (i.e. split up in most cases) those original regions so that the correlation coefficients are higher than 0.7 between all HEGIFTOM sites within a region. As a consequence, each region contains only well-correlated sites in terms of TrOC/FTOC. By adding some sites with different co-located techniques (Mauna Loa and Lauder) and the Canadian Arctic to the original regional selection in Gaudel et al. (2020) and Wang et al. (2022), we end up with 24 well-correlated regions that are covered with ground-based HEGIFTOM data.

We tried to clarify this better in the paper. First, at the end of the first paragraph of Sect. 3.1, we changed the sentence into "We use the 11 regions defined in Gaudel et al. (2020) and Wang et al. (2022) on the basis of IAGOS and ozonesonde profile measurements as the starting points for merging time series, but the final regional domains are refined based on the spatial correlation characteristics."

Then, the description of Fig. 1 and the second paragraph in Sect. 3.1. now reads as "As can be seen in Fig 1a, all of continental Europe seems to be well-correlated in terms of tropospheric ozone column monthly anomalies, while a further subdivision of East Asia, one of the regions defined in Wang et al. (2022), their Fig. 1, is required to define well-correlated regions (see Fig. 1b). In the latter case, the East Asia region shown in Fig. 1a was divided into three different well-correlated regions (e.g. regions 9, 10, and 11 in Fig. 3). In general, this correlation analysis method resulted in the refinement (i.e. splitting up) of many of the 11 original regions defined by Gaudel et al. (2020) and Wang et al. (2022): Europe (in Continental Europe, European Arctic), Western North America (Western USA, Pacific Northwest, California), East Asia (East China, Northeast Asia, South Japan), Southeast Asia (Southeast Asia, South China Sea), Malaysia/Indonesia (Indonesia, Southern Malay Peninsula), Persian Gulf (Persian Gulf, East Mediterranean Sea), Gulf of Guinea (Gulf of Guinea, West African Highlands), Northern South America (Middle America, Caribbean). The remaining three regions (Eastern North America, Southeast US, India) in Gaudel et al (2020) and Wang et al. (2022) did not need to be divided because all of their sites were well-correlated in terms of TrOC/FTOC. By adding some sites with different co-located techniques (Mauna Loa and Lauder) and the Canadian Arctic to the original regions defined by Gaudel et al. (2020) and Wang et al. (2022), we end up with 24 well-correlated regions that are covered by ground-based HEGIFTOM data. Those regions form the starting point for the regional trend estimations, for both TOST and the synthetized trend approaches, but the spatial and temporal sampling of the ground-based observations in those regions place further constraints on the final determination and selection of the regions."

And in Figure 1, I suggest showing the locations of IAGOS FRA airport and the FTIR Hefei station.

Done. Thank you very much for this good suggestion.

3) Page 10, Figure 3 –

Figure 3 shows the locations of 19 different regions. Were these regions identified by the spatial correlation analysis (which said 24 different regions)? Please clarify.

Yes. The 24 different regions were computed, as explained above, based on the correlation coefficients between the CAMS TrOC/FTOC at the HEGIFTOM site locations. However, the ground-based measurements themselves contain gaps. Only measurement sites with time series having at least 30 months of data were included in the regions, and only regions consisting of at least 2 sites fulfilling this data coverage criterion. In the end, 5 of those 24 regions therefore had to be discarded from our analysis.

In the text, we first specified that "In practice, the LMM method will be applied on the L1 (all measurements) data of the sites within one of the 24 regions defined in Sect. 3.1". Then, we further wrote that "Because we included only sites with observation time series covering at least 30 months in the well-correlated regions, and we consider only regions consisting of at least 2 sites fulfilling this criterion, 5 of those 24 regions had to be discarded from the LMM trend estimation. The sites retained in each preserved region are listed in Tables 1 and S1, and shown in Fig. 3."

4) Page 14, Line 350-357 -

For the TOST-based trends, 12 regions were selected and analyzed, as shown in Figure 4. It was not clear why the 19 regions as defined in Figure 3 were not used to facilitate the comparisons between TOST-based and HEGIFTOM-based trends. Please clarify.

The definition of the TOST regions was also based on the 24 regions from the correlation analysis, but, in contrast with the sample of sites used for the correlation analysis, TOST only deals with ozonesonde sites, not with IAGOS, FTIR, Umkehr, and lidar sites. So, the TOST sample to be used for defining regions is more limited, e.g. discerning sites with different techniques co-located, and moreover, the density of the TOST measurements (or the number of independent samples, see Figs. 4c and 4d) further limits the number of regions for the TOST trend estimations.

The text at the end of Sect. 4 has been modified to "Because the number of independent samples is (too) low in about half of the 24 well-correlated regions determined by the correlation analysis (see Section 3.1), we could select only 12 regions (Fig. 4a and Table 3) for analyzing TOST-based TrOC and FTOC trends. Within each region, high spatial correlation (>0.7, in line with Weatherhead et al., 2017) is found among stations. Table 3 lists the column-averaged mean independent samples over 1995-2021 in those 12 regions. The number and spatial extent of some of those regions is different from the (19) regions used for calculating LMM synthesized trends (see Sect. 3.2, Fig. 3 and Table 2), because these latter regions include sites or airports where ozone is measured with techniques or platforms other than ozonesondes (i.e. FTIR, IAGOS, Umkehr, and lidar)."

5) Figures 10-13

Figures 10-13 appear to need large revisions. The font size was difficult to read, and the font type was different from that of other figures, such as Figure 9.

Thank you. We revised Figures 10-13, increasing the font sizes, symbols, and thickness of the lines in the graphs. All figures in the manuscript now also use the same font type, Helvetica.

6) Since several regression methods were utilized in the study, a Table or some paragraphs summarizing their applications and differences would be helpful to smooth the paper structure. Right now, it is not clear in the manuscript why Quantile Regression was used in one case, while Dynamical Linear Modelling was used in another case. Would these regression methods return different findings?

Very good idea to provide such a table. We generalized the idea and created a table summarizing the main differences between the TOST and LMM approach, not only in terms of the statistics used, but also included the measurements used, the time coverage, how the regions have been defined, the merging method, etc. We hope that this table might also clarify some of the issues that you raised in earlier points.

This table can be found in the beginning of Sect. 5 (Trends) and the following text has been added to introduce it:

"In this section, we will present and discuss the trend estimates for the different well-correlated regions, optimized based on sampling density for each of the (gridded) TOST and (individual) HEGIFTOM ground-based sites. The specific properties of these two datasets calls for the use of different statistical trend estimation methods for determining regional trends: the TOAR-II recommended (Chang et al., 2023b) Quantile Regression (QR) and DLM (for a subset only) are applied to the merged gridded TOST data within a region, whereas the LMM utilizes a linear regression model to synthetize (or merge) the trends from the individual site time series in a region. The main

characteristics of both datasets and the different statistical techniques are summarized in Table 4, which serves as the framework for this section.

Table 4. Summary of the most important differences between the two approaches that are used to calculate regional trends from the TOST and HEGIFTOM datasets.

| | TOST | LMM |
|---|---|---|
| Measurements | Ozonesondes only, trajectory mapped | IAGOS, homogenized ozonesondes, FTIR, Umkehr, Lidar (HEGIFTOM) |
| Time coverage | 1990-2021, 1995-2021, 2000-2021 | 1990-2022, 1995-2022, 2000-2022 |
| Regions | N = 12 (see Fig. 4a and Table 3), based on correlation analysis (3.1) and density of independent samples (Figs. 4c & d) | N = 19 (see Fig. 3 and Table 1), based on correlation analysis (3.1) and availability of $\geq 2$ sites with $\geq 30$ months of data. |
| Merging method | Merging of gridded trajectory-mapped data and ozonesonde **measurements** (if available) within a region | Synthetizing **trends** from individual sites within a region |
| Trend estimation tools | QR trend estimation from the merged, regional, annual mean time series. DLM (Sect. 3.3) is used on a subset of regions using monthly mean (L3) time series (Fig. 8), refining the decadal trend estimation by showing how the trend varies over time | Linear regression model for calculating synthetized trends from well-correlated individual time series (L1, all measurements), allowing an intercept and a slope to adjust the difference from each individual trend against the overall trends (Sect. 3.2) |

We also specifically added to the DLM description in Sect. 3.3: "DLM is only applied to three TOST regional monthly mean time series, to show how the trends for these regions changed over time." From the yearly DLM trends, the decadal trends over the considered time periods 1995-2021 and 2000-2021 might be derived. However, in Van Malderen et al. (2025), we already showed that the DLM and QR trend estimates for the individual site time series were very similar. This was already included in Sect. 3.3.

7) Page 20, Figure 8

The legends p300 and FT need to be defined in the figure caption. DLM trends were presented here and in the Section 5.1.2, right? I do not see anywhere else described the DLM results (not in Sect 5.3 as said on Line 378).

We changed the legends to TrOC and FTOC, the acronyms that have been used throughout the manuscript to respectively denote the tropospheric ozone column between the surface to 300 hPa and the free-tropospheric ozone column between 700 and 300 hPa. DLM results are indeed only shown in Figure 8 and described in Section 5.1.2., while the DLM method itself is (briefly) described in Sect. 3.3 (and not 5.3) and referenced there for further details (Ball et al., 2017, Alsing et al., 2019).

8) Page 30, Line 667

"The difference is, at least for some regions, driven by the positive trends from measurement techniques other than ozonesondes." What did the statement mean by trends from measurement techniques? Please better explain it.

TOST trends are only based on ozonesonde data, while the LMM method calculates synthetized trends from ozonesonde, FTIR, lidar, IAGOS, and Umkehr time series. If some of the densely sampled techniques (FTIR, lidar, Umkehr) display a strong (positive) trend in a region, the LMM trend is expected to be impacted considerably w.r.t. the TOST trend for the same region, without including data from those techniques. In the discussion of Sect. 5.3, some examples are given for the Europe and North America (except California) regions.

We have changed this sentence to "The difference is, at least for some regions, driven by the positive trends from the measurement techniques other than ozonesondes (FTIR, Umkehr, IAGOS, lidar) that have been included in LMM, but not in TOST."

9) "TO BE FURTHER COMPLETED" in the Acknowledgements shall be noted.

Thank you, we completed the acknowledgements!

---

## Author Comment (AC2)

Response to Referee #2 comment on "Ground-based Tropospheric Ozone Measurements: Regional tropospheric ozone column trends from the TOAR-II/ HEGIFTOM homogenized datasets" by Van Malderen et al.

This paper presents a careful quantification of regional trends in tropospheric ozone from a range of ground-based measurements. The paper is well written, the analysis is thorough and the results are a valuable addition to the field.

My comments are minor. See below.

Thanks you for your minor comments. We believe we have answered and implemented those and thank you for your help in improving the manuscript.

**General comments**

I didn't catch why the chosen time periods run through 2021 for the TOST analysis but through 2022 for the HEGIFTOM analysis. I would suggest that the authors make it clear why a different date range was chosen for TOST vs HEGIFTOM.

The TOST dataset, developed by Liu et al., (2013a, b), has recently been improved and updated up to 2021 by Zang et al. (2024). After 2021, it is not yet available.

According to the TOAR-II Community Special Issue Guidelines (https://igacproject.org/sites/default/files/2023-04/TOAR-II_Community_Special_Issue_Guidelines_202304.pdf), 21st Century trends are defined for "time series beginning in the range 2000-2002 and ending in the range 2019-2021 (may include sites with data before 2000, but limit the analysis to 2000 and later)". So, strictly speaking, we could have limited all the HEGIFTOM time series up to 2021 as well, but we preferred to add the extra year of data, if available, also for the calculating the trends from the individual site time series (accompanying paper Van Malderen et al., 2025).

At the end of Sect. 5.1, we added the sentence "It should be noted that all trends are calculated up to 2021, as the most updated version of the TOST dataset (Zang et al., 2024) is provided until the end of 2021."

I did not follow the reasoning as to why different statistical approaches were applied to the TOST vs the HEGIFTOM datasets. It would be helpful to include some more explanation for the rationale for the choice of these particular approaches for application to these particular datasets.

We included a new Table 4 in the manuscript (see in our response to Referee #1 to find the new table) that summarizes the most important differences between the TOST and LMM approaches, not only in terms of the statistics. The choice of the statistical approach is tied to the characteristic of each dataset: TOST provides a gridded, trajectory mapped, monthly mean (tropospheric) ozone dataset, whereas the HEGIFTOM dataset is made of (partial) tropospheric ozone column time series at individual sites, from different techniques. So, to obtain regional trends from TOST, trends are estimated from merged data with QR, while with the LMM approach, synthetized (linear regression) trends from (well-correlated) individual time series within a region are calculated.

**Specific comments**

Abstract: "challenged by the diversity between satellite tropospheric ozone records and the sparse temporal and spatial sampling of ground-based measurements".

I had to read this quite a few times before I could understand what this sentence was supposed to mean, particularly the phrase "challenged by the diversity". I think that the authors are pointing to two different issues. One issue is that there are numerous satellite products for tropospheric ozone and that there are differences between those records that have not yet been well understood or accounted for. The other issue is that the "reference" ground-based datasets have

limitations in their spatial and temporal coverage. I think it would be much better to split this into two sentences to make the points clear.

We changed this sentence into "Quantifying long-term free-tropospheric ozone trends is essential for understanding the impact of human activities and climate change on atmospheric chemistry. However, this task is complicated by two key challenges: the differences among existing satellite-derived tropospheric ozone products, which are not yet fully understood or reconciled, and the limited temporal and spatial coverage of ground-based reference measurements."

Line 87: "available satellite products disagreed on the sign of the trend". This may be overly picky, but I would say that the analysis in the Gaudel et al. (2018) paper was not sufficient to determine whether there was truly disagreement or whether the differences between the trends presented were due to inherent characteristics of the different satellite datasets. I would suggest saying something like "early analysis of available satellite products did not provide a consistent picture of the sign of the trend".

We implemented your suggestion.

Line 89 and Conclusions: Can you state at the end whether or not your conclusions are consistent with those from the IPCC AR6?

This is a very good point, and we made this comparison. We added the following bullet point to the Conclusions: "These findings are consistent with the conclusions of IPCC AR6. The HEGIFTOM regional trends in the free troposphere (FTOC, 700-300 hPa) and in the tropospheric column (TrOC, p< 300 hPa) for the period 1995-2019 (see Figure 12a and Tables S2 and S3), are very similar to the free tropospheric and tropospheric column trends assessed by IPCC AR6 (see Figure 2.8 in Gulev et al., 2021), which span a similar time period."

Line 122: It would be helpful to state in this paragraph what the 5 different measurement techniques are, since this isn't explained until later.

Done. Thank you for addressing this.

Section 2.2: Is it correct to say that there is no chemistry in TOST? Ozone is assumed to be constant along a given trajectory? Is that what "assigned along its forward and backward trajectory" means? Please clarify.

The answer is that TOST does not include photochemistry directly. However, TOST does account for chemistry to some extent. This is associated with the assumption that ozone concentrations in an air parcel remain constant along a 4-day trajectory (backward and forward) because the lifetime of ozone is generally longer than 4 days in the atmosphere. The forward or backward trajectory starts from the location of an ozonesonde at a given height for every 1 km from the surface to 26 km. The ozone mixing ratio value at that height is assigned to each grid cell along its 4-day forward and backward trajectories.

We included the sentence "As the lifetime of ozone is generally longer than 4 days in the atmosphere (Stevenson et al., 2006; Monks et al., 2015; Han et al., 2019; Prather and Zhu, 2024), it is conservative to assume that the ozone mixing ratio in an air parcel is constant along a given trajectory of 4 days, running either forward or backward from an ozonesonde profile." And further, we completed a sentence (italic) with "Rather than simple linear or polynomial interpolation, the ozone measurement from the ozonesonde profile at the origin of a trajectory is assigned along its forward and backward trajectory paths, *starting from a location of the ozonesonde at a given height for every 1 km from the surface to 26 km. The ozone mixing ratio value at that height is assigned to each grid cell along its 4-day forward and backward trajectories.*"

Han, H., Liu, J., Yuan, H., Wang, T., Zhuang, B., and Zhang, X.: Foreign influences on tropospheric ozone over East Asia through global atmospheric transport, Atmos. Chem. Phys., 19, 12495–12514, https://doi.org/10.5194/acp-19-12495-2019, 2019.

Monks, P. S., Archibald, A. T., Colette, A., Cooper, O., Coyle, M., Derwent, R., Fowler, D., Granier, C., Law, K. S., Mills, G. E., Stevenson, D. S., Tarasova, O., Thouret, V., von Schneidemesser, E., Sommariva, R., Wild, O., and Williams, M. L.: Tropospheric ozone and its precursors from the urban to the global scale from air quality to short-lived climate forcer, Atmos. Chem. Phys., 15, 8889–8973, https://doi.org/10.5194/acp-15-8889-2015, 2015.

Prather, M. J. and Zhu, X.: Lifetimes and timescales of tropospheric ozone: Global metrics for climate change, human health, and crop/ecosystem research, Elementa: Science of the Anthropocene, 12, 1, https://doi.org/10.1525/elementa.2023.00112, 2024.

Stevenson, D. S., Dentener, F. J., Schultz, M. G., Ellingsen, K., Van Noije, T. P. C., Wild, O., Zeng, G., Amann, M., Atherton, C. S., Bell, N., Bergmann, D. J., Bey, I., Butler, T., Cofala, J., Collins, W. J., Derwent, R. G., Doherty, R. M., Drevet, J., Eskes, H. J., Fiore, A. M., Gauss, M., Hauglustaine, D. A., Horowitz, L. W., Isaksen, I. S. A., Krol, M. C., Lamarque, J. F., Lawrence, M. G., Montanaro, V., Müller, J. F., Pitari, G., Prather, M. J., Pyle, J. A., Rast, S., Rodgriguez, J. M., Sanderson, M. G., Savage, N. H., Shindell, D. T., Strahan, S. E., Sudo, K. and Szopa, S.: Multimodel ensemble simulations of present-day and near-future tropospheric ozone. J. Geophys. Res. 111(D8). DOI: https://doi.org/10.1029/2005JD006338, 2006.

Caption for Figure 5: Figures show trends from 1995-2021 but statistics are based on median values over 1990-2021. Is this correct or should the date ranges be consistent?

In the caption of Figure 5, the 1990-2021 is a mistake. It should be 1995-2021. We are sorry for this. Thanks for pointing it out!

Line 543: There is a sentence that starts with "And". Please check grammar.

We changed this:
"Alternatively, the impact of the FTIR and Umkehr time series, if available, on the TrOC trend, could be rather limited."

Line 580: Consider adding a sentence or two that says something about possible reasons for free tropospheric increases that are not driven by precursor emissions/lower tropospheric increases. Changes in dynamics? Changes in OH availability?

We added: "While regional ozone trends can be influenced by interannual variability resulting from meteorological influences (e.g. ENSO) (Chandra et al., 1998; Oman et al., 2011, 2013; Ziemke et al., 2015; Lin et al., 2015, 2015, 2017; Lu et al., 2019; Xue et al., 2021; Jeong et al., 2023; Stauffer et al., 2024), model studies have consistently indicated that the hemispheric scale increase of ozone precursor emissions, especially in the tropics, is the dominant driver of positive ozone trends in the free troposphere of northern mid-latitudes (Verstraeten et al., 2015; Zhang et al., 2016, 2021; Fiore et al., 2022; Liu et al., 2022; Wang et al., 2022)."

Chandra, S., J. R. Ziemke, W. Min, and W. G. Read (1998), Effects of 1997–1998 El Niño on tropospheric ozone and water vapor, Geophys. Res. Lett., 25, 3867–3870.

Fiore, Arlene M., Sarah E. Hancock, Jean-François Lamarque, Gustavo P. Correa, Kai-Lan Chang, Muye Ru, Owen R. Cooper, Audrey Gaudel, Lorenzo M. Polvani, Bastien Sauvage and Jerry R. Ziemke (2022), Understanding recent tropospheric ozone trends in the context of large internal variability: A new perspective from chemistry-climate model ensembles, Environmental Research: Climate, https://doi.org/10.1088/2752-5295/ac9cc2

Jeong, Y., Kim, S.-W., Kim, J., Shin, D., Kim, J., Park, J.-H., & An, S.-I. (2023). Influence of ENSO on tropospheric ozone variability in East Asia. Journal of Geophysical Research: Atmospheres, 128, e2023JD038604. https://doi.org/10.1029/2023JD038604

Lin, M., Horowitz, L.W., Oltmans, S.J., Fiore, A.M. and Fan, S., 2014. Tropospheric ozone trends at Mauna Loa Observatory tied to decadal climate variability. Nature Geoscience, 7(2), pp.136-143

Lin, M., Fiore, A.M., Horowitz, L.W., Langford, A.O., Oltmans, S.J., Tarasick, D. and Rieder, H.E., 2015. Climate variability modulates western US ozone air quality in spring via deep stratospheric intrusions. Nature communications, 6(1), p.7105.

Lin, M., et al. (2017), US surface ozone trends and extremes from 1980 to 2014: quantifying the roles of rising Asian emissions, domestic controls, wildfires, and climate, Atmos. Chem. Phys., 17, 2943–2970, 2017, www.atmos-chem-phys.net/17/2943/2017/doi:10.5194/acp-17-2943-2017

Liu, J., Strode, S. A., Liang, Q., Oman, L. D., Colarco, P. R., Fleming, E. L., et al. (2022). Change in tropospheric ozone in the recent decades and its contribution to global total ozone. Journal of Geophysical Research: Atmospheres, 127, e2022JD037170. https://doi.org/10.1029/2022JD037170

Lu X, Zhang L, Zhao Y, et al. Surface and tropospheric ozone trends in the Southern Hemisphere since 1990: possible linkages to poleward expansion of the Hadley Circulation. Sci Bull 2019; 64:400–9.

Oman, L. D., J. R. Ziemke, A. R. Douglass, D. W. Waugh, C. Lang, J. M. Rodriguez, J. E. Nielsen (2011), The response of tropical tropospheric ozone to ENSO, Geophys. Res. Lett., 38, doi:10.1029/2011GL047865

Oman et al. (2013), The ozone response to ENSO in Aura satellite measurements and a chemistry-climate simulation, JOURNAL OF GEOPHYSICAL RESEARCH: ATMOSPHERES, VOL. 118, 965–976, doi:10.1029/2012JD018546, 2013

Wang, H., Lu, X., Jacob, D. J., Cooper, O. R., Chang, K.-L., Li, K., Gao, M., Liu, Y., Sheng, B., Wu, K., Wu, T., Zhang, J., Sauvage, B., Nédélec, P., Blot, R., and Fan, S. (2022a), Global tropospheric ozone trends, attributions, and radiative impacts in 1995–2017: an integrated analysis using aircraft (IAGOS) observations, ozonesonde, and multi-decadal chemical model simulations, Atmos. Chem. Phys., 22, 13753–13782, https://doi.org/10.5194/acp-22-13753-2022

Xue, L., Ding, A., Cooper, O., Huang, X., Wang, W., Zhou, D., Wu, Z., McClure-Begley, A., Petropavlovskikh, I., Andreae, M.O. and Fu, C., 2021. ENSO and Southeast Asian biomass burning modulate subtropical trans-Pacific ozone transport. National Science Review, 8(6), p.nwaa132.

Zhang, Y., West, J. J., Emmons, L. K., Flemming, J., Jonson, J. E., Lund, M. T., et al. (2021). Contributions of World Regions to the Global Tropospheric Ozone Burden Change from 1980 to 2010. Geophysical Research Letters, 48, e2020GL089184. https://doi.org/10.1029/2020GL089184

Ziemke et al. (2015), Tropospheric ozone variability in the tropics from ENSO to MJO and shorter timescales, Atmos. Chem. Phys., 15, 8037–8049, www.atmos-chem-phys.net/15/8037/2015/

Caption for Figure 13: Please state what the gray points represent. I see that this is stated in the text, but it should also be made clear in the figure caption.

It was already there: "In grey: the individual site trend estimates *from Van Malderen et al. (2025)*, with different symbols for the different techniques", but we added, in italic, the reference to the HEGIFTOM individual site trends paper.

---

## Author Comment (AC3)

Response to Gabriele Pfister (TOAR Steering Committee Member) comment on "Ground-based Tropospheric Ozone Measurements: Regional tropospheric ozone column trends from the TOAR-II/ HEGIFTOM homogenized datasets" by Van Malderen et al.

This review is by Gabriele Pfister, member of the TOAR-II Steering Committee. The primary purpose of these reviews is to identify any discrepancies across the TOAR-II submissions, and to allow the author teams time to address the discrepancies. Additional comments may be included with the reviews. While members of the TOAR Steering Committee may post open comments on papers submitted to the TOAR-II Community Special Issue, they are not involved with the decision to accept or reject a paper for publication, which is entirely handled by the journal's editorial team.

**Comments regarding TOAR-II guidelines:**

TOAR-II has produced two guidance documents to help authors develop their manuscripts so that results can be consistently compared across the wide range of studies that will be written for the TOARII Community Special Issue. Both guidance documents can be found on the TOAR-II webpage: https://igacproject.org/activities/TOAR/TOAR-II. The TOAR-II Community Special Issue Guidelines: In the spirit of collaboration and to allow TOAR-II findings to be directly comparable across publications, the TOAR-II Steering Committee has issued this set of guidelines regarding style, units, plotting scales, regional and tropospheric column comparisons, and tropopause definitions.

The TOAR-II Recommendations for Statistical Analyses: The aim of this guidance note is to provide recommendations on best statistical practices and to ensure consistent communication of statistical analysis and associated uncertainty across TOAR publications. The scope includes approaches for reporting trends, a discussion of strengths and weaknesses of commonly used techniques, and calibrated language for the communication of uncertainty. Table 3 of the TOAR-II statistical guidelines provides calibrated language for describing trends and uncertainty, similar to the approach of IPCC, which allows trends to be discussed without having to use the problematic expression, "statistically significant".

**General Comments**

The study uses the homogenized ground-based and in-situ tropospheric ozone measurements from the TOAR-II Focus Working Group HEGIFTOM (Harmonization and Evaluation of Ground-based Instruments for Free-Tropospheric Ozone Measurements) to conduct an in-depth analysis of regional trends of total and partial tropospheric ozone columns over the 1995-2022 time period. It builds on the previous Van Malderen et al. (2024) analysis which presented trends for the individual sites based on the HEGIFTOM data set.

The overarching objective of TOAR-II is to provide an updated state of science of the ozone global distribution and trends relevant to climate, human health and vegetation. As such this paper makes an important contribution to the TOAR-II objective and also provides valuable input for evaluating chemical transport models as well as guidance on calculating trends.

The study is very detailed and carefully conducted and limitations in both the measurements and statistical analysis are characterized. The authors introduce new approaches to define regions for the trend calculation (a re-analysis based correlation analysis), include the Trajectory-mapped Ozonesonde dataset for the Stratosphere and Troposphere (TOST) in their analysis for comparison and present a linear mixed-effects modelling approach to calculate synthesized trends. Where possible, the authors compare their calculated trends to previous studies and if discrepancies arise, they are investigated and if possible explained.

While the trends in many parts could be explained by changing precursor emissions, I think the results do give motivation for further studies to understand what other drivers besides emissions might be contributing to the trends.

Thank you for your careful assessment of our study, Gabi.

**Specific Comments**

The paper requires careful reading because of the amount of detail discussed and the different advanced statistical methods involved. It would help for understanding to make more clear what the different methods were used for and why the different methods, specifically as it relates to LMM and DLM. Otherwise readers who are not fully familiar with advanced statistics might find it challenging to follow the paper.

This is really a common comment raised by all the reviewers. So, we included a table that summarizes the most important differences between the LMM and TOST approach, and why different statistical tools are used for either the TOST or the HEGIFTOM dataset. In the response to Referee #1, you will find the new Table 4.

The figures contain a lot of information and this makes them hard to read. Maybe there is a way to simplify some of them and keep the current complex figures in a supplement.

We are aware that the figures contain a lot of information. We invest quite some time in making them clearer in terms of font/symbol sizes, line thickness, etc. and to make them more consistent between the different techniques.

E.g.,: Figure 6 & 7: It is not clear to me what the difference between filled and open symbols is. Maybe it would help to discern the different regions (which I found challenging because some of the colors look very similar) and the region-specific colors and symbols could be used for all the plots in a consistent manner (at the moment squares are used for median trends and circles for the difference plots).

We did not use filled and open symbols in those plots anymore. We also consistently use the same colours and figures for similar regions for TOST and LMM in the manuscript's figures now.

Figure 8: Increase the font size for the labels

Done. Thank you for pointing this out.

Figure 10&11&12: Maybe increasing the font size and making the symbols larger could help with the readability of the graphs?

Done. Thank you!

Figure 3: It might be of interest to also include sites that were not used in the analysis in a different symbol.

On Figure 3, we only included the sites that actually have been used in the analysis, which complies with the listing in Table 1. The figure caption now reads as "Map showing the 19 different regions (squares, numbered) and the individual HEGIFTOM ground-based sites (black dots) that are used for the regional synthetized trends calculation using the linear mixed-effects modelling (LMM) approach. An overview of those different regions and included HEGIFTOM sites is given in Tables 1 and S1."